# Large Language Models (LLMs) on Tabular Data: Prediction, Generation, and Understanding - A Survey

**Xi Fang***  *lxifan@amazon.com*
*Amazon*

**Weijie Xu***  *weijiexu@amazon.com*
*Amazon*

**Fiona Anting Tan***†  *fiona.anting.tan@gmail.com*
*National University of Singapore*

**Jiani Zhang**  *zhajiani@amazon.com*
*AWS*

**Ziqing Hu**  *ziqinghu@amazon.com*
*AWS*

**Yanjun Qi**  *yanjunqi@amazon.com*
*AWS*
*University of Virginia*

**Scott Nickleach**  *nickleac@amazon.com*
*Amazon*

**Diego Socolinsky**  *sclinsky@amazon.com*
*AWS*

**Srinivasan Sengamedu**  *sengamed@amazon.com*
*Amazon*

**Christos Faloutsos**  *faloutso@amazon.com*
*Carnegie Mellon University*
*Amazon*

**Reviewed on OpenReview:** *https://openreview.net/forum?id=IZnrCGF9WI&noteId=nWxFR4OvnD*

## Abstract

Recent breakthroughs in large language modeling have facilitated rigorous exploration of their application in diverse tasks related to tabular data modeling, such as prediction, tabular data synthesis, question answering, and table understanding. Each task presents unique challenges and opportunities. However, there is currently a lack of comprehensive review that summarizes and compares the key techniques, metrics, datasets, models, and optimization approaches in this research domain. This survey aims to address this gap by consolidating recent progress in these areas, offering a thorough survey and taxonomy of the datasets, metrics, and methodologies utilized. It identifies strengths, limitations, unexplored territories, and gaps in the existing literature, while providing some insights for future research directions in this vital and rapidly evolving field. It also provides relevant code and datasets references. Through this comprehensive review, we hope to provide interested readers with

---

*These authors contributed equally to this work.
†The author worked on this project during her intern at Amazon.

pertinent references and insightful perspectives, empowering them with the necessary tools and knowledge to effectively navigate and address the prevailing challenges in the field.

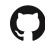 https://github.com/tanfiona/LLM-on-Tabular-Data-Prediction-Table-Understanding-Data-Generation

# 1 Introduction

Large language models (LLMs) are deep learning models trained on extensive data, endowing them with versatile problem-solving capabilities that extend far beyond the realm of natural language processing (NLP) tasks (Fu & Khot, 2022). Recent research has revealed emergent abilities of LLMs, such as improved performance on few-shot prompted tasks (Wei et al., 2022b). The remarkable performance of LLMs have incited interest in both academia and industry, raising beliefs that they could serve as the foundation for Artificial General Intelligence (AGI) of this era (Chang et al., 2024; Zhao et al., 2023b; Wei et al., 2022b). A noteworthy example is ChatGPT, designed specifically for engaging in human conversation, that demonstrates the ability to comprehend and generate human language text (Liu et al., 2023g).[1]

Before LLMs, researchers have been investigating ways to integrate tabular data with neural network for NLP and data management tasks (Badaro et al., 2023). Today, researchers are keen to investigate the abilities of LLMs when working with tabular data for various tasks, such as prediction, table understanding, quantitative reasoning, and data generation (Hegselmann et al., 2023; Sui et al., 2023c; Borisov et al., 2023a).

Tabular data stands as one of the pervasive and essential data formats in machine learning (ML) (van Breugel & van der Schaar, 2024), with widespread applications across diverse domains such as finance, medicine, business, agriculture, education, and other sectors that heavily rely on relational databases (Sahakyan et al., 2021; Rundo et al., 2019; Hernandez et al., 2022; Umer et al., 2019; Luan & Tsai, 2021).

In the current work, we provide a comprehensive review of recent advancements in modeling tabular data using LLMs. In the first section, we introduce the characteristics of tabular data, then provide a brief review of traditional, deep-learning and LLM methods tailored for this area. In Section 2, we introduce key techniques related to the adaptation of tabular data for LLMs. Subsequently, we cover the applications of LLMs in prediction tasks (Section 3), data augmentation and enrichment tasks (Section 4), and question answering/table understanding tasks (Section 5). Finally, Section 6 discusses limitations and future directions, while Section 7 concludes. The overview of this paper is shown in Figure 1.

## 1.1 Characteristics of tabular data

Tabular data, commonly known as structured data, refers to data organized into rows and columns, where each column represents a specific feature. This subsection discusses the common characteristics and inherited challenges with tabular data:

1. Heterogeneity: Tabular data can contain different feature types: categorical, numerical, binary, and textual. Therefore, features can range from being dense numerical features to sparse or high-cardinality categorical features (Borisov et al., 2022a).

2. Sparsity: Real-world applications, such as clinical trials, epidemiological research, fraud detection, etc., often deal with imbalanced class labels and missing values, which results in long-tailed distribution in the training samples (Sauber-Cole & Khoshgoftaar, 2022).

3. Dependency on pre-processing: Data pre-processing is crucial and application-dependent when working with tabular data. For numerical values, common techniques include data normalization or scaling, categorical value encoding, missing value imputation, and outlier removal. For categorical values, common techniques include label encoding or one-hot encoding. Improper pre-processing may lead to information loss, a sparse matrix, and it may introduce multi-collinearity (e.g. with

---

[1]We would like to thank Fanyou for his valuable contributions in discussing the project and idetifying relevant methods.

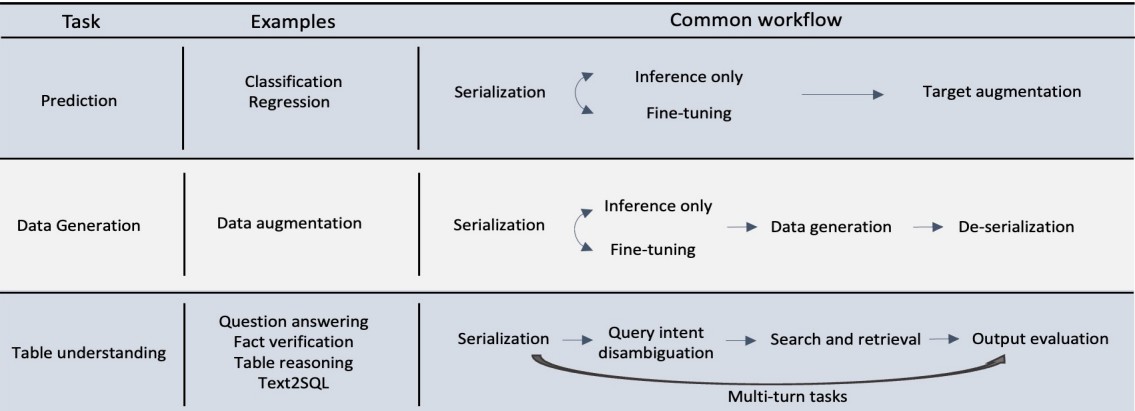

Figure 1: Overview of LLM on Tabular Data: the paper discusses application of LLM for prediction, data generation, and table understanding tasks

one-hot encoding), or it may introduce synthetic ordering (e.g. with ordinal encoding) (Borisov et al., 2023a).

4. Context-based interconnection: In tabular data, features can be correlated. For example, age, education, and alcohol consumption from a demographic table are interconnected: it is hard to get a doctoral degree at a young age, and there is a minimum legal drinking age. Including correlated regressors in regressions lead to biased coefficients, hence, a modeler must be aware of such intricacies (Liu et al., 2023d).

5. Order invariant: In tabular data, samples and features can be sorted. However, as opposed to text-based and image-based data that is intrinsically tied to the position of the word/token or pixel in the text or image, tabular data are relatively order-invariant. Therefore, position-based methodologies (e.g., spatial correlation, impeding inductive bias, convolutional neural networks (CNN)) are less applicable for tabular data modeling (Borisov et al., 2022a).

6. Lack of prior knowledge: In image or audio data, there is often prior knowledge about the spatial or temporal structure of the data, which can be leveraged by the model during training. However, in tabular data, such prior knowledge is often lacking, making it challenging for the model to understand the inherent relationships between features (Borisov et al., 2022a; 2023a).

## 1.2 Traditional and deep learning in tabular data

This survey explores the current research landscape of LLMs in tabular data prediction, with a focus on classification task, data generation, and table understanding.

Tabular prediction refers to classification and regression tasks. For tabular prediction, despite advancements in the field, traditional tree-based ensemble methods such as gradient-boosted decision trees (GBDT) remain the state-of-the-art (SOTA) for classification task on tabular data (Borisov et al., 2022a; Gorishniy et al., 2021)). In boosting ensemble methods, base learners are learned sequentially to reduce previous learner's error until there is no significant improvement, making it relatively stable and accurate than a single learner (Chen & Guestrin, 2016). Traditional tree-based models are known for its high performance, efficiency in training, ease of tuning, and ease of interpretation. However, they have limitations compared to deep learning models: 1. Tree-based models can be sensitive to feature engineering especially with categorical features while deep learning can learn representation implicitly during training (Goodfellow et al., 2016). 2. Tree-based models are not naturally suited for processing sequential data, such as time series while deep learning models such as Recurrent Neural Networks (RNNs) and transformers excel in handling sequential

dependencies. 3. Tree-based models sometimes struggle to generalize to unseen data particularly if the training data is not representative of the entire distribution, while deep learning methods may generalize better to diverse datasets with their ability to learn intricate representations (Goodfellow et al., 2016).

For deep learning methods in tabular data prediction, the methodologies can be broadly grouped into the following categories: 1. Data transformation. These models either strive to convert heterogenous tabular input into homogenous data more suitable to neural networks, like an image, on which CNN-like mechanism can be applied (SuperTML (Sun et al., 2019), IGTD (Zhu et al., 2021b), 1D-CNN (Kiranyaz et al., 2019)), or methods focusing on combining feature transformation with deep neural networks (Wide&Deep (Cheng et al., 2016; Guo & Berkhahn, 2016), DeepFM (Guo et al., 2017), DNN2LR (Liu et al., 2021)). 2. Differentiable trees. Inspired by the performance of ensembled trees, this line of methods seeks to make trees differentiable by smoothing the decision function (NODE (Popov et al., 2019), SDTR (Luo et al., 2021), Net-DNF (Katzir et al., 2020)). Another subcategory of methods combine tree-based models with deep neural networks, thus can maintain tree's capabilities on handling sparse categorical features (Deep-GBM (Ke et al., 2019a)), borrow prior structural knowledge from the tree (TabNN (Ke et al., 2019b)), or exploit topological information by converting structured data into a directed graph (BGNN (Ivanov & Prokhorenkova, 2021). 3. Attention-based methods. These models incorporate attention mechanisms for feature selection and reasoning (TabNet (Arik & Pfister, 2020)), feature encoding (TransTab (Wang & Sun, 2022), TabTransformer (Huang et al., 2020), TP-BERTa (Yan et al., 2024b)), feature interaction modeling (DANet (Chen et al., 2022a), T2G-Former (Yan et al., 2023), ExcelFormer (Chen et al., 2023a), ARM-net (Cai et al., 2021)), or aiding intrasample information sharing (SAINT (Somepalli et al., 2021), NPT (Kossen et al., 2022)). 4. Regularization methods. The importance of features varies in tabular data, in contrast to image or text data. Thus, this line of research seeks to design an optimal and dynamic regularization mechanism to adjust the sensitivity of the model to certain inputs (e.g. RLN (Shavitt & Segal, 2018), Regularization Cocktails (Kadra et al., 2021). In spite of rigorous attempts in applying deep learning to tabular data modeling, GBDT algorithms, including XGBoost, LightGBM, and CatBoost (Prokhorenkova et al., 2019), still outperform deep-learning methods in most datasets with additional benefits in fast training time, high interpretability, and easy optimization (Shwartz-Ziv & Armon, 2022; Gorishniy et al., 2021; Grinsztajn et al., 2022). Deep learning models, however, may have their advantages over traditional methods in some circumstances, for example, when facing very large datasets, or when the data is primarily comprised of categorical features (Borisov et al., 2022a).

Another important task for tabular data modeling is data synthesis. The ability to synthesize real and high-quality data is essential for model development. Data generation is used for augmentation when the data is sparse (Onishi & Meguro, 2023), imputing missing values (Jolicoeur-Martineau et al., 2023), and class rebalancing in imbalanced data (Sauber-Cole & Khoshgoftaar, 2022). Traditional methods for synthetic data generation are mostly based on Copulas (Patki et al., 2016; Li et al., 2020b) and Bayesian networks (Zhang et al., 2017; Madl et al., 2023) while recent advancement in generative models such as Variational Autoencoders (VAEs) (Ma et al., 2020; Darabi & Elor, 2021; Vardhan & Kok, 2020; Liu et al., 2023d; Xu et al., 2023b)), generative adversarial networks (GANs) (Park et al., 2018; Choi et al., 2018; Baowaly et al., 2019; Xu et al., 2019), diffusion models(Kotelnikov et al., 2022; Xu et al., 2023a; Kim et al., 2022b;a; Lee et al., 2023; Zhang et al., 2023c), and LLMs, opened up many new opportunities. These deep learning approaches have demonstrated superior performance over classical methods such as Bayesian networks ((Xu et al., 2019)). A comprehensive understanding of the strengths and weaknesses of different tabular data synthesis methods can be found in Du & Li (2024).

Table understanding is a broad field, covering various tasks like question answering (QA), natural language inference (NLI), Text2SQL tasks, and more. Many earlier methods fine-tune BERT (Devlin et al., 2019) to become table encoders for table-related tasks, like TAPAS (Herzig et al., 2020), TABERT (Yin et al., 2020a), TURL (Deng et al., 2022a), TUTA (Wang et al., 2021) and TABBIE (Iida et al., 2021). For example, TAPAS extended BERT's masked language model objective to structured data by incorporating additional embeddings designed to capture tabular structure. It also integrates two classification layers to facilitate the selection of cells and predict the corresponding aggregation operator. A particular table QA task, Text2SQL, involves translating natural language question into structured query language (SQL). Earlier research conducted semantic parsing through hand-crafted features and grammar rules (Pasupat

& Liang, 2015b). Semantic parsing is also used when the table is not coming from non-database tables such as web tables, spreadsheet tables, and others (Jin et al., 2022). Seq2SQL is a sequence-to-sequence deep neural network using reinforcement-learning to generate conditions of query on WikiSQL task (Zhong et al., 2017a). Some methodologies are sketch-based, wherein a natural language question is translated into a sketch. Subsequently, programming language techniques such as type-directed sketch completion and automatic repair are utilized in an iterative manner to refine the initial sketch, ultimately producing the final query (e.g. SQLizer (Yaghmazadeh et al., 2017)). Another example is SQLNet (Xu et al., 2017) which uses column attention mechanism to synthesize the query based on a dependency graph-dependent sketch. A derivative of SQLNet is TYPESQL (Yu et al., 2018a) which is also a sketch-based and slot-filling method entails extracting essential features to populate their respective slots. Unlike the previous supervised end-to-end models, TableQuery is a NL2SQL model pretrained on QA on free text that obviates the necessity of loading the entire dataset into memory and serializing databases.

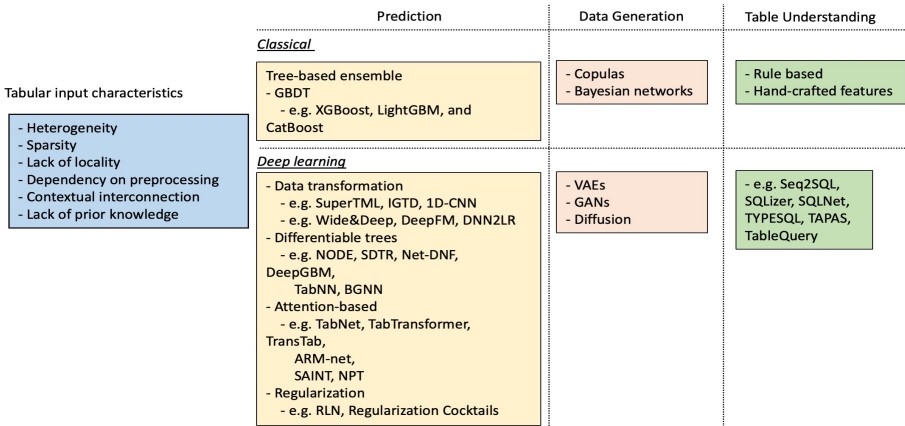

Figure 2: Tabular data characteristics and machine learning models for tabular data prediction, data synthesis and table understanding like question answering before LLMs.

### 1.3 Overview of large language models (LLMs)

A language model (LM) is a probabilistic model that predicts the generative likelihood of future or missing tokens in a word sequence. Zhao et al. (2023b) thoroughly reviewed the development of LMs, and characterized the it into four different stages: The first stage is **Statistical Language Models (SLM)**, which learns the probability of word occurrence in an example sequence from previous words (e.g. N-Gram) based on Markov assumption (Saul & Pereira, 1997). Although a more accurate prediction can be achieved by increasing the context window, SLM is limited by the curse of high dimensionality and high demand for computation power (Bengio et al., 2000). Next, **Neural Language Models (NLM)** utilize neural networks (e.g. Recurrent neural networks (RNN)) as a probabilistic classifier (Kim et al., 2015). In addition to learning the probabilistic function for word sequence, a key advantage of NLM is that they can learn the distributed representation (i.e. word embedding) of each word so that similar words are mapped close to each other in the embedding space (e.g. Word2Vec); thus, the model can generalize well to unseen sequences, as well as it avoids the curse of dimensionality (Bengio et al., 2000). Later, rather than learning a static word embedding, context-aware representation learning was introduced by pretraining the model on large-scale unannotated corpora using bidirectional LSTM that takes context into consideration (e.g., ELMo (Peters et al., 2018a)), which shows significant performance boost in various natural language processing (NLP) tasks (Wang et al., 2022a; Peters et al., 2018b). Along this line, several other **Pretrained Language Models (PLM)** were proposed utilizing a transformer architecture with self-attention mechanisms including BERT and GPT2 (Ding et al., 2023). The pre-training and fine-tuning paradigm, closely related to transfer learning, allows the model to gain general syntactic and semantic understanding of the text corpus and then be trained on task-specific objectives to adapt to various tasks. The final and most recent stage of LM is the **Large Language Models (LLMs)**, and will be the focus of this paper. Motivated by the observation that scaling

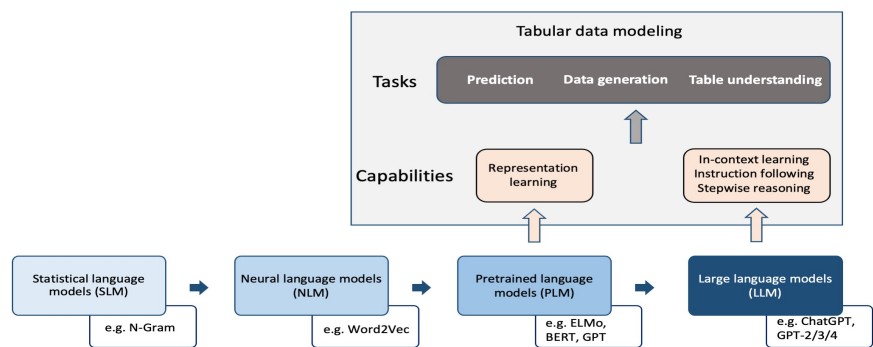

Figure 3: Development of language models and their applications in tabular data modeling.

the data and model size usually leads to improved performance, researchers sought to test the boundaries of PLM's performance of a larger size, such as "text-to-text transfer transformers" (T5) (Raffel et al., 2023), GPT-3 (Brown et al., 2020), etc. Intriguingly, some advanced abilities emerge as a result. These large-sized PLMs (i.e. LLMs) show unprecedentedly powerful capabilities (also called emergent abilities) that go beyond traditional language modeling and start to gain capability to solve more general and complex tasks which was not seen in PLM. Formally, we define a LLM as follows:

**Definition 1** (Large Language Model). A large language model (LLM) $M$, parameterized by $\theta$, is a transformer-based model with an architecture that can be autoregressive, autoencoding, or encoder-decoder. It has been trained on a large corpus comprising hundreds of millions to trillions of tokens. LLMs encompass pre-trained models and for our survey, refers to models that have at least 1 billion parameters.

Several key emergent abilities of LLMs are critical for data understanding and modeling including **in-context learning**, **instruction following**, and **multi-step reasoning**. In-context learning refers to designing large auto-regressive language models that generate responses on unseen task without gradient update, only learning through a natural language task description and a few in-context examples provided in the prompt. The GPT3 model (Brown et al., 2020) with 175 billion parameters presented an impressive in-context learning ability that was not seen in smaller models. LLMs have also demonstrated the ability to complete new tasks by following only the instructions of the task descriptions (also known as zero-shot prompts). Some papers also fine-tuned LLMs on a variety of tasks presented as instructions (Thoppilan et al., 2022). However, instruction-tuning is reported to work best only for larger-size models (Wei et al., 2022a; Chung et al., 2022). Solving complex tasks involving multiple steps have been challenging for LLMs. By including intermediate reasoning steps, prompting strategies such as chain-of-thought (CoT) has been shown to help unlock the LLM ability to tackle complex arithmetic, commonsense, and symbolic reasoning tasks (Wei et al., 2023). These new abilities of LLMs lay the groundwork for exploring their integration into intricate tasks extending beyond traditional NLP applications across diverse data types.

### 1.3.1 Applications of LLMs in tabular data

Despite the impressive capabilities of LM in addressing NLP tasks, its utilization for tabular data learning has been constrained by differences in the inherent data structure. Some research efforts have sought to utilize the generic semantic knowledge contained in PLM, predominantly BERT-based models, for modeling tabular data (Figure 3). This involves employing PLM to learn contextual representation with semantic information taking header information into account (Chen et al., 2020c). The typical approach includes transforming tabular data into text through serialization (detailed explanation in Section 2) and employing a masked-language-modeling (MLM) approach for fine-tuning the PLM, similar to that in BERT (PTab, CT-BERT, TABERT (Liu et al., 2022a; Ye et al., 2023a; Yin et al., 2020a). In addition to being able to incorporate semantic knowledge from column names, converting heterogenous tabular data into textual

representation enables PLMs to accept inputs from diverse tables, thus enabling cross-table training. Also, due to the lack of locality in tabular data, models need to be invariant to permutations of the columns (Ye et al., 2023a). In this fashion, TABERT was proposed as a PLM trained on both natural language sentence and structured data (Yin et al., 2020a), PTab demonstrated the importance of cross-table training for an enhanced representation learning (Liu et al., 2022a), CT-BERT employs masked table modeling (MTM) and contrastive learning for cross-table pretraining that outperformed tree-based models (Ye et al., 2023a). However, previous research primarily focuses on using LM for representation learning, which is quite limited.

### 1.3.2 Opportunities for LLMs in tabular data modeling

Many studies today explore the potential of using LLMs for various tabular data tasks, ranging from prediction, data generation, to data understanding (further divided into question answering and data reasoning). This exploration is driven by LLMs' unique capabilities such as in-context learning, instruction following, and step-wise reasoning. The opportunities for applying LLMs to tabular data modeling are as follows:

1. Deep learning methods often exhibit suboptimal performance on datasets they were not initially trained on, making transfer learning using the pre-training and fine-tuning paradigm highly promising (Shwartz-Ziv & Armon, 2022).

2. The transformation of tabular data into LLM-readable natural language addresses the curse of dimensionality (created by the one-hot encoding of categorical data).

3. The emergent capabilities, such as step-by-step reasoning through CoT, have transformed LM from language modeling to a more general task-solving tool. Research is needed to test the limit of LLM's emergent abilities on tabular data modeling.

### 1.4 Contribution

The key contributions of this work are as follows:

1. **A formal break down of key techniques for LLMs' applications on tabular data** We split the application of LLM in tabular data to tabular data prediction, tabular data synthesis, tabular data question answering and table understanding. We further extract key techniques that can apply to all applications. We organize these key techniques in a taxonomy that researchers and practitioners can leverage to describe their methods, find relevant techniques and understand the difference between these techniques. We further subdivide each technique to subsections so that researchers can easily find relevant benchmark techniques and properly categorize their proposed techniques.

2. **A survey and taxonomy of metrics for LLMs' applications on tabular data.** For each application, we categorize and discuss a wide range of metrics that can be used to evaluate the performance of that application. For each application, we documented the metric of all relevant methods, and we identify benefits/limitations of each class of metrics to capture application's performance. We also provide recommended metrics when necessary.

3. **A survey and taxonomy of datasets for LLMs' applications on tabular data.** For each application, we identify datasets that are commonly used for benchmark. For table understanding and question answering, we further categorize datasets by their downstream applications: Question Answering, Natural Language Generation, Classification, Natural Language Inference and Text2SQL. We further provided recommended datasets based on tasks and their GitHub link. Practitioners and researchers can look at the section and find relevant dataset easily. We share publicly-available datasets here: https://github.com/tanfiona/LLM-on-Tabular-Data-Prediction-Table-Understanding-Data-Generation

4. **A survey and taxonomy of techniques for LLMs' applications on tabular data.** For each application, we break down an extensive range of tabular data modeling methods by steps. For

example, tabular data prediction can be breakdown by pre-processing (modifying model inputs), target augmentation (modifying the outputs), fine-tuning (fine-tuning the model). We construct granular subcategories at each stage to draw similarities and trends between classes of methods, and we provide examples of main techniques. Practitioners and researchers can look at the section and understand the difference of each technique. We only recommend benchmark methods and provide GitHub link of these techniques for reference and benchmark.

5. **An overview of future research directions.** Future research could focus on how to solve bias problem in tabular data modeling, how to mitigate hallucinations, how to find better representations of numerical data, how to improve capacity, how to form standard benchmarks, how to improve model interpretability, how to create an integrated workflow, how to design better fine-tuning strategies and finally, how to improve the performance of downstream applications.

## 2 Key techniques for LLMs' applications on tabular data

While conducting our survey, we noticed a few common components in modeling tabular data with LLMs across tasks. We discuss common techniques, like serialization, table manipulations, prompt engineering, and building end-to-end systems in this section. Fine-tuning LLMs is also popular, but it tends to be application-specific, thus we discuss it later, in Sections 3 and 5.

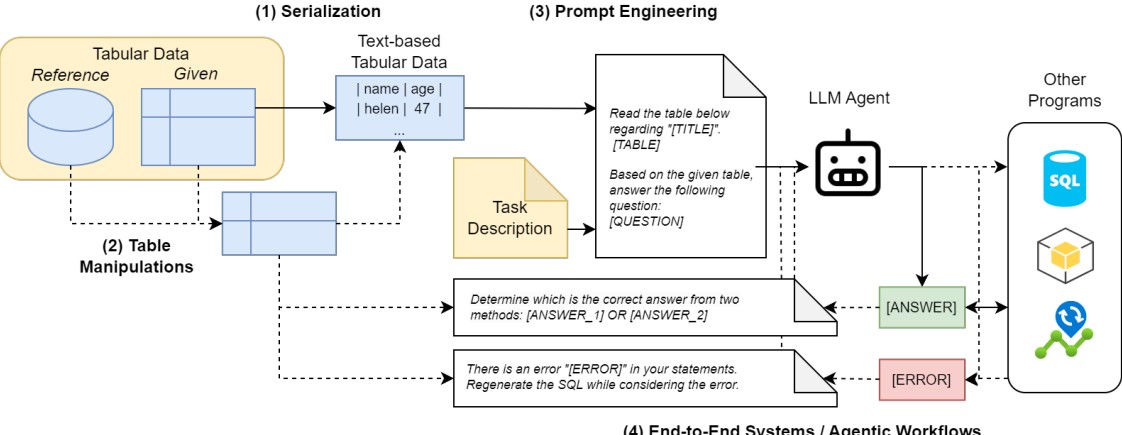

Figure 4: Key techniques in using LLMs for tabular data. The dotted line indicates steps that are optional.

### 2.1 Serialization

Since LLMs are sequence-to-sequence models, in order to feed tabular data as inputs into an LLM, we have to convert the structured tabular data into a text format (Sui et al., 2023b; Jaitly et al., 2023).

**Text-based** Table 1 describes the common text-based serialization methods in the literature. A straightforward way would be to directly input a programming-language readable data structure (E.g. Pandas DataFrame Loader for Python, line-separated JSON-file format, Data Matrix represented by a list of lists, HTML code reflecting tables, etc). Alternatively, the table could be converted into X-separated values, where X could be any reasonable delimiter like comma or tab. Some papers convert the tables into human-readable sentences using templates based on the column headers and cell values. The most common approach based on our survey is the Markdown format.

**Embedding-based** Many papers also employ table encoders, which were fine-tuned from PLMs, to encode tabular data into numerical representations as the input for LLMs. There are multiple table encoders, built on BERT (Devlin et al., 2019) for table-related task, like TAPAS (Herzig et al., 2020), TABERT (Yin et al.,

| Method | Description | Example | Papers that investigated this |
|---|---|---|---|
| DFLoader | Python code where a dictionary is loaded as a Pandas dataframe | `pd.DataFrame({` `name:['helen'], age:[47] })` | Singha et al. (2023) |
| JSON | Row number as indexes, with each row represented as a dictionary of keys (column names) and values | `{"0": {"name": "helen", "age": "47"}}` | Singha et al. (2023); Sui et al. (2023b) |
| Data Matrix | Dataframe as a list of lists, where the firm item is the column header | `[['','name','age']` `[0, 'helen', 47]]` | Singha et al. (2023) |
| Markdown | Rows are line-separated, columns are separated by "\|" [2] | `\|  \| name \| age \|` `\|:--\|:-----\|----:\|` `\|0  \|helen \|   47\|` | Singha et al. (2023); Liu et al. (2023e); Zhang et al. (2023d); Ye et al. (2023b); Zhao et al. (2023d); Sui et al. (2023b) |
| LaTeX | Rows are separated by "\\\hline", columns are separated by "&" | `\\\hline helen & 47` | Jaitly et al. (2023) |
| X-Separated | Rows are line-separated, columns are separated by ",", "\t", ":", etc. | `, name, age` `0, helen, 47` | Singha et al. (2023); Narayan et al. (2022) |
| Attribute-Value Pairs | Concatenation of paired columns and cells {c : v} | `name:helen ; age:47` | Wang et al. (2023c) |
| HTML | HTML element for tabular data | `<table><thead><tr><th></th>` `<th>name</th><th>age</th></tr>` `</thead><tbody><tr><th>0</th>` `<td>helen</td><td>47</td></tr>` `</tbody></table>` | Singha et al. (2023); Sui et al. (2023c;b) |
| Sentences | Rows are converted into sentences using templates | `name is helen, age is 47` | Yu et al. (2023); Hegselmann et al. (2023); Gong et al. (2020); Dinh et al. (2022); Jaitly et al. (2023) |

Table 1: Text-based serialization methods.

2020b), TURL (Deng et al., 2022a), TUTA (Wang et al., 2021), TABBIE (Iida et al., 2021) and UTP (Chen et al., 2023b). Cong et al. (2023) discuss the pros and cons of the learned table representations of a few of these encoders. For LLMs with >1B parameters, there are UniTabPT (Sarkar & Lausen, 2023) with 3B parameters (based on T5 and Flan-T5 models)), TableGPT (Gong et al., 2020) with 1.5B parameters (based on GPT2), TableGPT[3] (Zha et al., 2023) with 7B parameters (based on Phoenix (Chen et al., 2023c)), TableLlama (Zhang et al., 2023f) with 7B parameters (based on Llama 2 (Touvron et al., 2023b)), and Table-GPT with 350M, 3B, 13B or 175B parameters (based on various versions of OpenAI's GPT models).

**Graph-based & Tree-based** A possible, but less explored serialization method involves converting a table to a graph or tree data structure. However, when working with sequence-to-sequence models, these structures must still be converted back to text. For Zhao et al. (2023a), after converting the table into a tree, each cell's hierarchical structure, position information, and content was represented as a tuple and fed into GPT3.5.

**Comparisons** Research has shown that LLM performance is sensitive to the input tabular formats. Singha et al. (2023) found that DFLoader and JSON formats are better for fact-finding and table transformation tasks. Meanwhile, Sui et al. (2023a) found that HTML or XML table formats are better understood by GPT models over tabular QA and FV tasks. However, they require increased token consumption. Likewise, Sui et al. (2023b) also found that markup languages, specifically HTML, outperformed X-separated formats for GPT3.5 and GPT4. Their hypothesis is that the GPT models were trained on a significant amount of web data and thus, probably exposed the LLMs to more HTML and XML formats when interpreting tables.

Apart from manual templates, Hegselmann et al. (2023) also used LLMs (Fine-tuned BLOOM on ToTTo (Parikh et al., 2020b), T0++ (Sanh et al., 2022), GPT-3 (Ouyang et al., 2022)) to generate descriptions of a table as sentences, blurring the line between a text-based and embedding-based serialization methodology.

---

[3]Same name, different group of authors.

However, for the few-shot classification task, they find that traditional list and text templates outperformed the LLM-based serialization method. Amongst LLMs, the more complex and larger the LLM, the better the performance (GPT-3 has 175B, T0 11B, and fine-tuned BLOOM model 0.56B parameters). A key reason why the LLMs are worse off at serializing tables to sentences is due to the tendency for LLMs to hallucinate: LLMs respond with unrelated expressions, adding new data, or return incorrect feature values.

## 2.2 Table Manipulations

Table manipulations refer to operations and transformations performed on tabular data, typically stored in databases or spreadsheets. These manipulations involve actions such as filtering, sorting, joining, aggregating, and transforming data. An important characteristic of tabular data is its heterogeneity in structure and content. They often come in large size with different dimensions encompassing various feature types. In order for LLMs to ingest tabular data efficiently, it is important to compress the tables to fit context lengths, for better performance and reduced costs. Therefore, table manipulations are required in some scenarios, as described below.

**Compacting tables to fit context lengths, for better performance and reduced costs** For smaller tables, it might be possible to include the whole table within a prompt. However, for larger tables, there are three challenges:

Firstly, some models have short context lengths (E.g. Flan-UL2 (Tay et al., 2023b) supports 2048 tokens, Llama 2 (Touvron et al., 2023b) supports 4096 context tokens) and even models that support large context lengths might still be insufficient for extremely large tables with over 200K rows (Claude 2.1 supports up to 200K tokens).

Secondly, even if the table could fit the context length, most LLMs are slow to process long sentences due to the quadratic complexity of self-attention (Sui et al., 2023b; Tay et al., 2023a; Vaswani et al., 2017). When dealing with long contexts, performance of LLMs significantly degrades when models must access relevant information in the middle of long contexts, even for explicitly long-context models (Liu et al., 2023b). For tabular data, Cheng et al. (2023); Sui et al. (2023c) highlights that noisy information becomes an issue in large tables for LMs. Chen (2023) found that for table sizes beyond 1000 tokens, GPT-3's performance degrades to random guesses.

Thirdly, longer prompts incur higher costs, especially for applications built upon LLM APIs.

To address these issues, Herzig et al. (2020); Liu et al. (2022c) proposed methods to truncate the input based on a maximum sequence length. Sui et al. (2023b) introduced predefined certain constraints to meet the LLM call request. Another strategy is to do search and retrieval of only highly relevant tables, rows, columns or cells which we will discuss later in Section 5.

**Additional information about tables for better performance** Apart from the table, some papers explored including table schemas and statistics as part of the prompt. Sui et al. (2023c) explored including additional information about the tables: Information like " dimension, measure, semantic field type" help the LLM achieve higher accuracy across all six datasets explored. "Statistics features" improved performance for tasks and datasets that include a higher proportion of statistical cell contents, like FEVEROUS (Aly et al., 2021). Meanwhile, "document references" and "term explanations" add context and semantic meaning to the tables. "Table size" had minimal improvements, while "header hierarchy" added unnecessary complexity, and hurt performance. For the Text2SQL task, Chang & Fosler-Lussier (2023) also find that some table relationships and database content are useful. Huang et al. (2023b) report improvements in GPT-4's accuracy by 28.9% when incorporating documentation that disambiguate terms present in the table like data column names, value consistency, data coverage, and granularity.

**Robustness of LLM performance to table manipulations** Liu et al. (2023e) critically analyzed the robustness of GPT3.5 across structural perturbations in tables (transpose and shuffle). They find that LLMs suffer from structural bias in the interpretation of table orientations, and when tasked to transpose the table, LLMs performs poorly ( 50% accuracy). However, LLMs can identify if the first row or first column

is the header (94-97% accuracy). Zhao et al. (2023e) investigated the effects of SOTA Table QA models on manipulations on the table header, table content and natural language question (phrasing).[4] They find that all examined Table QA models (TaPas, TableFormer, TaPEX, OmniTab, GPT3) are not robust under adversarial attacks.

## 2.3 Prompt Engineering

A prompt is an input text that is fed into an LLM. Designing an effective prompt is a non-trivial task, and many research topics have branched out from prompt engineering alone. In this subsection, we cover the popular techniques in prompt engineering, and how researchers have used them for tasks involving tables.

**Prompt format**   The simplest format is concatenating task description with the serialized table as string. An LLM would then attempt to perform the task described and return a text-based answer. Clearly-defined and well-formatted task descriptions are reported to be effective prompts (Marvin et al., 2023). Some other strategies to improve performance are described in the next few paragraphs. Sui et al. (2023b) recommended that external information (such as questions and statements) should be placed before the tables in prompts for better performance.

**In-context learning**   As one of the emergent abilities of LLMs (see 1.3), in-context learning refers to incorporates similar examples to help the LLMs understand the desired output. Sui et al. (2023b) observed significant performance drops performance, of overall accuracy decrease of 30.38% on all tasks, when changing their prompts from a 1-shot to a 0-shot setting. In terms of choosing appropriate examples, Narayan et al. (2022) found their manually curated examples to outperform randomly selected examples by an average of 14.7 F1 points. For Chen (2023), increasing from 1-shot to 2-shot can often benefit the model, however, further increases did not help.

**Chain-of-Thought and Self-consistency**   Chain-of-Thought (CoT) (Wei et al., 2022c) induces LLMs to decompose a task by performing step-by-step thinking, resulting in better reasoning. Program-of-Thoughts (Chen et al., 2022b) guides the LLMs using code-related comments like "*Let's write a program step-by-step...*". Zhao et al. (2023d) explored CoT and PoT strategies for the numerical QA task. Yang et al. (2023) prompt the LLMs with one shot CoT demonstration example to generate a reasoning and answer. Subsequently, they included the reasoning texts, indicated by special "<CoT>" token, as part of inputs to fine-tune smaller models to generate the final answer.

Self-consistency (SC) (Wang et al., 2023b) leverages the intuition that a complex reasoning problem typically admits multiple different ways of thinking leading to its unique correct answer. SC samples a diverse set of reasoning paths from an LLM, then selects the most consistent answer by marginalizing out the sampled reasoning paths. Inspired by these strategies, Zhao et al. (2023a); Ye et al. (2023b) experimented with multi-turn dialogue strategies, where they decompose the original question into sub-tasks or sub-questions to guide the LLM's reasoning. Sui et al. (2023c) instructed the LLM to "*identify critical values and ranges of the last table related to the statement*" to obtain additional information that were fed to the final LLM, obtaining increased scores for five datasets. Liu et al. (2023e) also investigated strategies around SC, along with self-evaluation, which guides the LLM to choose between the two reasoning approaches based on the question's nature and each answer's clarity. Deng et al. (2022b) did consensus voting across a sample a set of candidate sequences, then selected final response by ensembling the derived response based on plurality voting.

Chen (2023) investigated the effects of both CoT and SC on QA and FV tasks. When investigating the explainability of LLM's predictions, Dinh et al. (2022) experimented with a multi-turn approach of asking GPT3 to explain its own prediction from the previous round, and guided the explanation response using CoT by adding the line "*Let's think logically. This is because*".

---

[4]For table headers, they explored synonym and abbreviation replacement perturbations. For table content, they explored five perturbations: (1) row shuffling, (2) column shuffling, (3) extending column names content into semantically equivalent expressions, (4) masking correlated columns (E.g. "Ranking" and "Total Points" can be inferred from one another), and (5) introducing new columns that are derived from existing columns. For the question itself, they perturbed questions at the word-level or sentence-level.

**Retrieval-augmented generation (RAG)** Retrieval-augmented generation (RAG) relies on the intuition that the LLMs are general models, but can be guided to a domain-specific answer if the user includes the relevant context within the prompts. By incorporating tables as part of the prompts, most papers described in this survey can be attributed as RAG systems. A particular challenge in RAG is to extract the most relevant information out of a large pool of data to better inform the LLMs. This challenge overlaps slightly with the strategies about table sampling mentioned earlier under Section 2.2. Apart from the aforementioned methods, Sundar & Heck (2023) designed a dual-encoder-based Dense Table Retrieval (DTR) model to rank cells of the table according to their relevance to the query. The ranked knowledge sources are incorporated within the prompt, and led to top ROUGE scores.

**Role-play** Another popular prompt engineering technique is role-play, which refers to including descriptions in the prompt about the person the LLM should portray as it completes a task. For example, Zhao et al. (2023a) experimented with the prompt "*Suppose you are an expert in statistical analysis.*".

## 2.4 End-to-end systems

Since LLMs can generate any text-based output, apart from generating human-readable responses, it could also generate code readable by other programs. Abraham et al. (2022) designed a model that converts natural language queries to structured queries, which can be run against a database or a spreadsheet. Liu et al. (2023e) designed a system where the LLM could interact with Python to execute commands, process data, and scrutinize results (within a Pandas DataFrame), iteratively over a maximum of five iterations. Zhang et al. (2023d) demonstrated that we can obtain errors from the SQL tool to be fed back to the LLMs. By implementing this iterative process of calling LLMs, they improved the success rate of the SQL query generation. Finally, Liu et al. (2023c) proposes a no-code data analytics platform that uses LLMs to generate data summaries, including generating pertinent questions required for analysis, and queries into the data parser. A survey by Zhang et al. (2023h) covers further concepts about natural language interfaces for tabular data querying and visualization, diving deeper into recent advancements in Text-to-SQL and Text-to-Vis domains.

## 3 LLMs for predictions

Several studies have leveraged LLMs for prediction in tabular data. This section will delve into the existing methodologies and advancements in two categories of tabular data: standard feature-based tabular data and time series data. Time series prediction differs from normal feature-based tabular data since the predictive power heavily relies on the past. For each category, we divide it into different steps which include preprocessing, fine-tuning, and target augmentation. Preprocessing explains how different prediction methods generate input to the language model. Preprocessing includes serialization, table manipulation, and prompt engineering. Target augmentation maps the textual output from LLMs to a target label for prediction tasks. At the end, we will briefly cover domain-specific prediction methods using LLMs.

## 3.1 Datasets

For task-specific fine-tuning, most datasets used for the prediction task are from UCI ML, OpenML, or a combo of 9 datasets created by Manikandan et al. (2023). Details of the datasets are in Table 2. OpenML has the highest number of datasets, but the size of the largest dataset is only 5600 rows. Half of the datasets in UCI ML collections are relevant to medical use cases. Thus, the combo of 9 datasets is recommended for benchmark [5] since it contains larger size datasets and more diverse feature sets. For general fine-tuning, published methods choose the Kaggle API[6] as it has 169 datasets, and its datasets are very diverse.

| Dataset | Dataset Number | Papers that used this dataset |
|---|---|---|
| OpenML | 11 | Dinh et al. (2022); Manikandan et al. (2023) |
| Kaggle API | 169 | Hegselmann et al. (2023); Wang et al. (2023a); Zhang et al. (2023a) |
| Combo | 9 | Hegselmann et al. (2023); Wang et al. (2023a); Zhang et al. (2023a) |
| UCI ML | 20 | Manikandan et al. (2023); Slack & Singh (2023) |
| DDX | 10 | Slack & Singh (2023) |

Table 2: Combo is the combination of the following dataset in the form of dataset name (number of rows, number of features): Bank (45,211 rows, 16 feats), Blood (748, 4), California (20,640, 8), Car (1,728, 8), Creditg (1,000, 20), Income (48,842, 14), and Jungle (44,819, 6), Diabetes (768, 8) and Heart (918, 11).

| Algorithm | Type | Method | Resource | Metric | Model Used |
|---|---|---|---|---|---|
| TabletSlack & Singh (2023) | Tabular | No Finetune | Low | F1 | GPTJ/Tk-Instruct/Flan T5 |
| SummaryBoostManikandan et al. (2023) | Tabular | No Finetune | High | RMSE | GPT3 |
| LIFTDinh et al. (2022) | Tabular | Finetune | High | MAE/RMSE | GPT3/GPTJ |
| TabLLMHegselmann et al. (2023) | Tabular | Finetune | High | AUC | GPT3/T0 |
| UnipredictWang et al. (2023a) | Tabular | Finetune | Low | ACC | GPT2 |
| GTLZhang et al. (2023a) | Tabular | Finetune | Low | ACC | LLaMA |
| SerializeLLMJaitly et al. (2023) | Tabular | Finetune | High | AUC | T0 |
| PromptCastXue & Salim (2022) | Time Series | Finetune | High | MAE/ RMSE/ Missing Rate | T5/Bigbird/LED |
| ZeroTSGruver et al. (2023) | Time Series | No Finetune | Low | MAE/ Scale MAE/ CRPS | GPT3/LLAMA2 |
| TESTSun et al. (2023a) | Time Series | Finetune | High | ACC/ RMSE | Bert/ GPT2/ ChatGLM/ LLaMa |
| TimeLLMJin et al. (2023a) | Time Series | Finetune | High | SMAPE/ MSAE/ OWA | LLAMA7B/ GPT2 |
| MediTabWang et al. (2023c) | Medical | Finetune | High | PRAUC/AUCROC | BioBert/GPT3.5/UnifiedQA-v2-T5 |
| CTRLLi et al. (2023e) | Finance | Finetune | High | AUC/LogLoss | Roberta/ChatGLM |
| FinPTYin et al. (2023) | CTR | Finetune | High | F1 Score | FlanT5/ChatGPT/GPT4 |

Table 3: Prediction methods. "Resource" is "high" if it has to finetune a model with size $\geq$ 1B even if it is PEFT. "Model Used" include all models used in the paper which includes serialization, preprocessing and model finetuning. ACC stands for accuracy. AUC stands for Area under the ROC Curve. MAE stands for mean absolute error. RMSE stands for root-mean-square error. F1 score is calculated from the precision and recall of the test, where the precision is the number of true positive results divided by the number of all samples predicted to be positive, including those not identified correctly, and the recall is the number of true positive results divided by the number of all samples that should have been identified as positive. CRPS is continous ranked probability score. We will introduce other metrics in relevant sections.

## 3.2 Tabular prediction

**Preprocessing** Preprocessing in LLM-based tabular prediction involves steps like table manipulation, serialization, and prompt engineering, which have been discussed earlier. Specifically, some LLM-based prediction methods incorporated a statistical summary of the tabular data as part of the input to LLM. Serialization in the prediction task is mostly Text-based (refer to Section 2.1). Prompt engineering includes incorporating task-specific cues and relevant samples into the prompt (refer to Section 2.1). The various preprocessing methods are illustrated in Table 4 and discussed in detail below.

As one of the earliest endeavors, LIFT (Dinh et al., 2022) tried a few different serialization methods, such as feature and value as a natural sentence such as "*The column name is Value*" or a set of equations, such as $col_1 = val_1, col_2 = val_2, ....$ The former achieves higher prediction accuracy, especially in low-dimensional tasks. The same conclusion was drawn in TabLLM (Hegselmann et al., 2023) where they evaluated 9 different serialization methods along with a description for the classification problem. They found that a textual enumeration of all features: 'The column name is Value', performs the best. For medical prediction, they mimic the thinking process of medical professionals as prompt engineering and found that LLM makes use of column name and their relationships in few-shot learning settings.

In a subsequent study, TABLET (Slack & Singh, 2023) included naturally occurring instructions along with examples for serialization. In this case, where the task is for medical diagnosis, naturally occurring instructions are from consumer-friendly sources, such as government health websites or technical references such as the Merck Manual. It includes instructions, examples, and test data points. They found that

---

[5]GitHub repository link `https://Github.com/clinicalml/TabLLM/tree/main/datasets`
[6]Link to the pre-trained data `https://Github.com/Kaggle/kaggle-api`

these instructions significantly enhance zero-shot F1 performance. However, experiments from TABLET revealed that LLMs tend to ignore instructions, even with examples, leading to prediction failures. Along this fashion, more studies tested a more complex serialization and prompt engineering method rather than a simple concatenation of feature and value for serialization.

The schema-based prompt engineering usually includes background information about the dataset, a task description, a summary, and example data points. Summary Boosting(Manikandan et al., 2023) serializes data and metadata into text prompts for summary generation. This includes categorizing numerical features and using a representative dataset subset selected via weighted stratified sampling based on language embeddings. Serialize-LM (Jaitly et al., 2023) introduces 3 novel serialization techniques that boost LLM performance in domain-specific datasets. They included related features in one sentence to make the prompt more descriptive and easier for an LLM to understand. Take the task of car classification as an example: attributes like make, color, and body type are now combined into a single,richer sentence. It leverages covariance to identify the most relevant features and either label them critically or add a sentence to explain the most important features. Finally, they converted tabular data into LaTeX code format. This LaTeX representation of the table was then used as the input for fine-tuning our LLM by just passing a row representation preceded by `hline` tag without any headers.

Another work worth mentioning is UniPredict (Wang et al., 2023a), which reformats metadata by consolidating arbitrary input M to a description of the target and the semantic descriptions of features. Feature serialization follows a "column name is value" format. The objective is to minimize the difference between the output sequence generated by the adapted LLM function and the reference output sequence generated from target augmentation (represented by serialized target). To make LLMs applicable to multiple tabular datasets at the same time, Generative Tabular Learning (GTL) was proposed by Zhang et al. (2023a). It includes two parts: 1) the first part specifies the task background and description with optionally some examples as in-context examples (Prompt Engineering); 2) the second part describes feature meanings and values of the current instance to be inferred (Serialization); LIFT and TabLLM have been benchmarked by at least 3 other papers. The code for both methods is available. [7]

Some other methods leverage an LLM to rewrite the serialization or perform prompt engineering. TabLLM (Hegselmann et al., 2023) showed that LLM is not good for serialization because it is not faithful and may hallucinate. Summary Boosting(Manikandan et al., 2023) uses GPT3 to convert metadata to data description and generate a summary for a subset of datasets in each sample round. TABLET (Slack & Singh, 2023) fits a simple model such as a one-layer rule set model or prototype with the 10 most important features on the task's full training data. It then serializes the logic into text using a template and revises the templates using GPT3. Based on their experiments, it was found that contrary to the naturally occurring instructions, LLM-generated instructions do not significantly improve performance.

**Target Augmentation** LLMs can solve complex tasks through text generation, however, the output is not always controllable (Dinh et al., 2022). As a result, mapping the textual output from LLMs to a target label for prediction tasks is essential. This is called target augmentation (Wang et al., 2023a). A straightforward but labor-intensive way is manual labeling, as was used by Serilize-LM (Jaitly et al., 2023). To be more automatic, LIFT (Dinh et al., 2022) utilizes $\#\#\#$ and $@@@$ to demarcate question-answer pairs and signify the end of generation. These markers prompt the LLM to position answers between $\#\#\#$ and $@@@$. This approach significantly aligns most generated answers with the intended labels. Additionally, to address potential inaccuracies in inference outputs, LIFT conducts five inference attempts, defaulting to the training set's average value if all attempts fail. In streamlining the two-step approach, TabLLM (Hegselmann et al., 2023) incorporates the use of Verbalizer (Cui et al., 2022) to map the answer to a valid class. To calculate AUCROC or AUCPR, the probability of the output is necessary. Thus, Verbalizer proves advantageous for closed-source models by enabling the assignment of output probability. UniPredict (Wang et al., 2023a) has the most complicated target augmentation. They transform the target label into a set of probabilities for each class via a function called "augment". Formally, for a target $T$ in an arbitrary dataset $D$, they define a function $augment(T) = (C, P)$, where $C$ are new categories of targets with semantic meaning and P are the assigned probabilities to each category. They extend the target into categorical one-hot encoding and then

---

[7]Here is the Github repo for TABLET `https://Github.com/dylan-slack/Tablet`, TabLLM `https://Github.com/clinicalml/TabLLM` and LIFT `https://Github.com/UW-Madison-Lee-Lab/LanguageInterfacedFineTuning`

use an external predictor to create the calibrated probability distributions. This replaces the 0/1 one-hot encoding while maintaining the final prediction outcome. Formally, given the target classes $t \in 0, ..., |C|$ and target probabilities $p \in P$, they define a function serialize target(t, p) that serializes target classes and probabilities into a sequence formatted as "class $t_1 : p_1, t_2 : p_2, . . . .$ ". We give an example for each method in 5

**Inference-Only Prediction** Some work uses LLMs directly for prediction without fine-tuning, we refer to these approaches as inference-only prediction. TABLET (Slack & Singh, 2023) utilizes models like Tk-Instruct (Wang et al., 2022b), Flan-T5 (Chung et al., 2022), GPT-J (Black et al., 2022), and ChatGPT to perform inference, but it reports that a KNN approach with feature weights from XGBoost surpasses Flan-T5 11b in performance using similar examples and instructions. Summary Boosting (Manikandan et al., 2023) creates multiple inputs through the serialization step. The AdaBoost algorithm then creates an ensemble of summary-based weak learners. While non-fine-tuned LLMs struggle with continuous attributes, summary boosting is effective with smaller datasets. Furthermore, its performance is enhanced using GPT-generated descriptions by leveraging existing model knowledge, underscoring the potential of LLMs in new domains with limited data. However, it does not perform well when there are many continuous variables. For any new LLM-based prediction method without any fine-tuning, we suggest benchmarking LIFT and TABLET. LIFT is the first LLM-based method for inference-only prediction. TABLET shows significantly better performance than LIFT with 3 different base models.

**Fine-tuning** For studies involving fine-tuning, they typically employ one of two distinct approaches. The first involves training an LLM model on large datasets to learn fundamental features before adapting it to specific prediction tasks. The second takes a pre-trained LLM and further trains it on a smaller, specific prediction dataset to specialize its knowledge and improve its performance on the prediction. LIFT (Dinh et al., 2022) fine-tunes pre-trained language models like GPT-3 and GPT-J using Low-Rank Adaptation (LoRA) on the training set. They found that LLM with general pretraining could improve performance. However, the performance of this method does not surpass the in-context learning result. TabLLM (Hegselmann et al., 2023) uses T0 model (Sanh et al., 2021) and T-few (Liu et al., 2022b) for fine-tuning. TabLLM has demonstrated remarkable few-shot learning capabilities, outperforming traditional deep-learning methods and gradient-boosted trees. TabLLM's efficacy is highlighted by its ability to leverage the extensive knowledge encoded in pre-trained LLMs from these models, requiring minimal labeled data. However, the sample efficiency of TabLLM is highly task-dependent. Other research also leverages T0 as the based model. Jaitly et al. (2023) uses T0 (Sanh et al., 2021). Compared to TabLLM, it is trained using Intrinsic Attention-based Prompt Tuning (IA3) (Liu et al., 2022b). However, this method only works for a few short learning, worse than baseline when the number of shots is more or equal to 128. T0 model (Sanh et al., 2021) is commonly used as the base model for tabular prediction fine-tuning.

PLM can be effectively adapted for diverse tabular prediction tasks, demonstrating their versatility across heterogeneous datasets (Yan et al., 2024b). UniPredict (Wang et al., 2023a) trains a single LLM (GPT2) on an aggregation of 169 tabular datasets with diverse targets and observes advantages over existing methods. This model does not require fine-tuning LLM on specific datasets. Model accuracy and ranking are better than XGBoost when the number of samples is small. The model with target augmentation performs noticeably better than the model without augmentation. It does not perform well when there are too many columns or fewer representative features. GTL (Zhang et al., 2023a) fine-tunes LLaMA to predict the next token using 115 tabular datasets. To balance the number of instances across different datasets, they randomly sample up to 2,048 instances from each tabular dataset for GTL. They employed GTL which significantly improves LLaMA in most zero-shot scenarios. Based on the current evidence, we believe that fine-tuning on large number of datasets could further improve the performance. However, both UniPredict and GTL have not released their code yet.

**Metric** Among all tabular prediction methods surveyed, AUC is mostly commonly used metric for classification prediction and RMSE is mostly commonly used metric for regression 3

| Methodology | Method | Example |
|---|---|---|
| Feature name + Feature Value + Predicted Feature Name | Dinh et al. (2022); Hegselmann et al. (2023) | Car Brand is Land Rover. Year is 2017. Repair claim is |
| Task Background + Feature meaning + Feature Value + Predicted Feature meaning | Zhang et al. (2023a) | The task is about fraud repair claim prediction. The brand of car is Land Rover. The produce year is 2017. The repair claim of the car is |
| Dataset Summary + LLM Processed Feature + Task | Manikandan et al. (2023) | Larger car is always more expensive. This is a 2017 Land Rover. Therefore, this car repair claim is (Fraudulent or Not Fraudulent): |
| Latex Format of features value + Task | Jaitly et al. (2023) | Is this car repair claim fraudulent? Yes or No? |
| Expert Task Understanding + Label + Task | Slack & Singh (2023) | Identify if the car repair claim is fraudulent. An older car is more likely to have a fraudulent repair claim. Features Car Brand: Land Rover Year: 2017. Answer with one of the following: Yes \| No |
| Dataset description + Feature meaning + Feature Value + Task | Wang et al. (2023a) | The dataset is about fraud repair claims. Car Brand is the brand of car. The year is the age when the car is produced. The features are: Car Brand is Land Rover. The year is 2017. Predict if this car repair claim is fraudulent by Yes for fraudulent, No for not fraudulent |

Table 4: Method and Example for different preprocessing for general predictive tasks. The example is to predict if a car repair claim is fraudulent or not.

### 3.3 Time Series Forecasting

Compared to prediction on feature-based tabular data with numerical and categorical features, time series prediction pays more attention to numerical features and temporal relations. Thus, serialization and target augmentation are more relevant to how to best represent numerical features. Many papers have claimed that they use LLM for time series. However, most of these papers use LLM which is smaller than 1B. We will not discuss these methods here. Please refer to Jin et al. (2023b) for a complete introduction of these methods.

**Preprocessing** PromptCast (Xue & Salim, 2022) uses raw value as input the time series data in text format and adds a minimal description of the task; as output, it uses the target after it converts it to a sentence. ZeroTS (Gruver et al., 2023) argues that numbers are not encoded well in the original LLM encoding method. Thus, it encodes numbers by breaking them down by a few digits or by each single digit for GPT-3 and LLaMA, respectively. It uses spaces and commas for separation and omitting decimal points. Time LLM (Jin et al., 2023a) involves concatenating time series sequences into embeddings and integrating them with word embeddings to create a comprehensive input. This input is complemented by dataset context, task instructions, and input statistics as a prefix. TEST (Sun et al., 2023a) introduces an embedding layer tailored for LLMs, using exponentially dilated causal convolution networks for time series processing. The embedding is generated through contrastive learning with unique positive pairs and aligning text and time series tokens using similarity measures. Serialization involves two QA templates, treating multivariate time series as univariate series for sequential template filling.

**Target Augmentation** In terms of output mapping, ZeroTS (Gruver et al., 2023) proposes drawing multiple samples and using statistical methods or quantiles for point estimates or ranges. For Time-LLM (Jin et al., 2023a), the output processing is done through flattening and linear projection. The target augmentation method of ZeroTS is easy to implement [8].

---

[8]The code is in `https://Github.com/ngruver/llmtime`

**Inference-Only Prediction** Similar to feature-based tabular prediction, researchers explored LLMs' performance for time series forecasting without fine-tuning. ZeroTS (Gruver et al., 2023) examines the use of LLMs like GPT-3 (Brown et al., 2020) and LLaMA-70B (Touvron et al., 2023a) directly for time series forecasting. It evaluates models using mean absolute error (MAE), Scale MAE, and continuous ranked probability score (CRPS), noting LLMs' preference for simple rule-based completions and their tendency towards repetition and capturing trends. The study reports that LLMs are able to capture time series data distributions and to handle missing data without special treatment. However, this approach is constrained by the size of the window and the tokenization method of numerical feature, preventing it from further improvement.

**Fine-tuning** Fine-tuning the model for time series prediction is more commonly seen in current research. PromptCast (Xue & Salim, 2022) tried the method of inference-only prediction or fine-tuning on task-specific datasets. It shows that larger models always perform better. Time LLM (Jin et al., 2023a) presents a novel approach to time series forecasting by fine-tuning LLMs like LLaMa (Touvron et al., 2023a) and GPT-2 (Brown et al., 2020). Time-LLM is evaluated using metrics like the symmetric mean absolute percentage error (SMAPE), the mean absolute scaled error (MSAE), and the overall weighted average (OWA). It demonstrates notable performance in few-shot learning scenarios, where only 5 percent or 10 percent of the data are used. This innovative technique underscores the versatility of LLMs in handling complex forecasting tasks. For TEST (Sun et al., 2023a), soft prompts are used for fine-tuning. The paper evaluates models like Bert, GPT-2 (Brown et al., 2020), ChatGLM (Zeng et al., 2023), and LLaMa Touvron et al. (2023a), using metrics like classification accuracy and RMSE. However, the result shows that this method is not as efficient and accurate as training a small task-oriented model. In general, currently, LLaMa is used as the base model by most papers we surveyed.

**Metric** MAE is the most common metric. Another popular metric is the Continuous Ranked Probability Score (CRPS) as it captures distributional qualities, allowing for comparison of models that generate samples without likelihoods. CRPS is considered an improvement over MAE as it does not ignore the structures in data like correlations between time steps. The Symmetric Mean Absolute Percentage Error (SMAPE) measures the accuracy based on percentage errors, the Mean Absolute Scaled Error (MASE) is a scale-independent error metric normalized by the in-sample mean absolute error of a naive benchmark model, and the Overall Weighted Average (OWA) is a combined metric that averages the ranks of SMAPE and MASE to compare the performance of different methods. Among those metrics, MAE and RMSE are used by at least half of our surveyed methods in time series.

| Method | Used Paper | Example |
|---|---|---|
| Adding Special Token before and after the answer | Dinh et al. (2022) | `### {Category} @@@` |
| Verbalizer | Hegselmann et al. (2023) | `Output -> {category1: probability1, .}` |
| Specific Prefix | Manikandan et al. (2023); Slack & Singh (2023) | Please answer with category 1, category 2, ... |
| Predict probability and recalibrate | Wang et al. (2023a) | `{category1: probability1}` => Calibrated by XGBoost |

Table 5: Target Augmentation methods, papers that used them, and examples

## 3.4 Applications of Prediction using LLM

**Medical Prediction** Medical data such as electronic health records (EHR) is a rich and complex source of information about patients' medical histories, treatments, and outcomes. It has more inherent complexity than simple tabular data. It captures information about patients' health over time, contains unstructured data such as clinical notes, has high interconnection between variables, contains missing data and noisy signals. The LM based model could capture the long-term dependencies among events such as diabetes and deal with unstructured data such as clinical notes. Thus, LM based models (McMaster et al., 2023; Steinberg et al., 2021; Rasmy et al., 2021; Li et al., 2020a) perform better than XGBoost. However, these models only focused on predicting a small fraction of the International Statistical Classification of Diseases and Related

Health Problems (ICD) codes. Currently, Meditab (Wang et al., 2023c) aims to create a foundation model in the medical field. For preprocessing, Meditab utilizes GPT-3.5 (Brown et al., 2020) to convert tabular data into textual format, with a focus on extracting key values. Subsequently, it employs techniques such as linearization, prompting, and sanity checks to ensure accuracy and to mitigate errors. For fine-tuning, the system further leverages multitask learning on domain-specific datasets, generates pseudo-labels for additional data, and refines them using Shapley scores. Pretraining on the refined dataset is followed by fine-tuning using the original data. The resulting model supports both zero-shot and few-shot learning for new datasets. GPT-3.5 accessed via OpenAI's API facilitates data consolidation and augmentation, while UnifiedQA-v2-T5 (Khashabi et al., 2022) is employed for sanity checks. Additionally, Meditab utilizes a pre-trained BioBert classifier (Lee et al., 2019). The system undergoes thorough evaluation across supervised, few-shot, and zero-shot learning scenarios within the medical domain, demonstrating superior performance compared to gradient-boosting methods and existing LLM-based approaches. However, it may have limited applicability beyond the medical domain. The code is available.[9] for tabular prediction tasks specifically in the medical domain. On top of AUCROC, they also use a precision-recall curve (PRAUC) for evaluation. PRAUC is useful in imbalanced datasets, which is always the case for medical data.

Without any pretraining, LLM has also demonstrated superior performance. CPLLM (Shoham & Rappoport, 2023) leverages LLMs (Llama2 and BioMedLM) and does fine-tuning with QLora to predict diseases using structured EHR data. CPLLM demonstrated significant improvements over the state-of-the-art in all tested disease prediction tasks. Additionally, this approach, with an extended sequence length, is also suitable for patients who were not hospitalized. LLM has also been combined with Vertical models to do medical prediction Yan et al. (2024a), showcasing remarkable performance even without any manual labels.

**Financial Prediction** FinPT (Yin et al., 2023) presents an LLM-based approach to financial risk prediction. The method involves filling tabular financial data into a pre-defined template, prompting LLMs like ChatGPT and GPT-4 to generate natural-language customer profiles. These profiles are then used to fine-tune large foundation models such as BERT (Devlin et al., 2019), employing the models' official tokenizers. The process enhances the ability of these models to predict financial risks, with Flan-T5 emerging as the most effective backbone model in this context, particularly across eight datasets. For financial data, FinBench contains 10 datasets with varied training set sizes (from 2k - 140k) and feature sizes (from 9 - 120) [10].

**Recommendation Prediction** CTRL (Li et al., 2023e) proposes a novel method for Click Through Rate (CTR) prediction by converting tabular data into text using human-designed prompts, making it understandable for language models. The model treats tabular data and generated textual data as separate modalities, feeding them into a collaborative CTR model and a pre-trained language model such as ChatGLM (Zeng et al., 2023), respectively. CTRL employs a two-stage training process: the first stage involves cross-modal contrastive learning for fine-grained knowledge alignment, while the second stage focuses on fine-tuning a lightweight collaborative model for downstream tasks. The approach outperforms all the SOTA baselines including semantic and collaborative models over three datasets by a significant margin, showing superior prediction capabilities and proving the effectiveness of the paradigm of combining collaborative and semantic signals. However, the code for this method is not available. They use LogLoss and AUC to evaluate the method.

## 4   LLMs for tabular data generation

Tabular data synthesis serves numerous purposes across diverse domains, including augmenting training datasets for machine learning models (Fonseca & Bacao, 2023) to improve models' predictive accuracy and generalization capabilities. Moreover, it's crucial for data privacy (Assefa et al., 2020), where it enables the creation of synthetic replicas of sensitive data, protecting confidential information while still preserving the statistical properties essential for analysis. Additionally, tabular data synthesis aids in data preprocessing, filling missing values (Zheng & Charoenphakdee, 2022) and ensuring dataset integrity and completeness. This enhances the reliability of subsequent analyses and model building.

---

[9] Available at `https://Github.com/RyanWangZf/MediTab`.

[10] The dataset is in `https://huggingface.co/datasets/yuweiyin/FinBench` and the code for FinPT is in `https://Github.com/YuweiYin/FinPT`

Recent studies have increasingly relied on LLMs to synthesize tabular data, leveraging their advanced generative capabilities developed through extensive training on vast text corpora, including markdown-formatted serialized tabular data. This proficiency allows LLMs to capture the intricate patterns and relationships inherent in tabular datasets (Bordt et al., 2024). Furthermore, LLMs possess rich language and data understanding, enabling them to produce synthetic datasets faithful to real-world statistics, with semantic coherence and contextuality (Sui et al., 2024).

## 4.1 Methodologies

Table 6 summarizes different LLM-powered table synthesis methods. Except for CLLM (Seedat et al., 2023), which utilizes prior knowledge from LLMs (e.g., GPT4) to augment and enhance training data samples in low-data settings without fine-tuning the LLMs, other methods such as GReaT (Borisov et al., 2023b), TAPTAP (Zhang et al., 2023e), TabuLa (Zhao et al., 2023f), and TabMT (Gulati & Roysdon, 2023) all involve fine-tuning the LLMs on a corresponding table. In standard data scenarios, fine-tuning an LLM to improve its ability to capture a table's data distribution becomes essential. This is because presenting the entire training table (often comprising millions of rows) to LLMs for in-context learning poses several challenges: 1) the low success ratio to extract the output cell values, where generated data samples may diverge from intended model output formats; 2) LLMs, acting as ICL, struggle to capture column-wise tail distributions due to the "lost-in-the-middle" phenomenon.

| | Used LLM | Fine-tuned or not | Serialization | Metric |
|---|---|---|---|---|
| GReaT (Borisov et al., 2023b) | GPT2/DistilGPT2 | Fine-tuned | Sentences | DCR, MLE |
| REaLTabFormer (Solatorio & Dupriez, 2023) | GPT2 | Fine-tuned | Sentences | DCR, MLE |
| TAPTAP (Zhang et al., 2023e) | GPT2/DistilGPT2 | Fine-tuned | Sentences | DCR, MLE |
| TabuLa (Zhao et al., 2023f) | DistilGPT2 | Fine-tuned | X-Separated | MLE |
| CLLM (Seedat et al., 2023) | GPT4 | Non Fine-tuned | X-Separated | MLE |
| TabMT (Gulati & Roysdon, 2023) | Masked Transformers -24layer | Fine-tuned | "[Value]" | MLE |

Table 6: LLM-powered data synthesis methods. "DCR" stands for Distance to the Closest Record and "MLE" stands for Machine Learning Efficiency.

In this section we survey methodologies that leverage LLMs for tabular data synthesis. We categorize the methods into two typical classes, Causal Language Modeling (CLM)-powered methods and Masked Language Modeling (MLM)-powered methods. CLM, as an autoregressive method used in GPT-based models, predicts the next token based on previous ones, focusing solely on past context. To model tabular data with unordered columns, permutation-invariant techniques are typically employed in CLM-powered methods. MLM involves masking tokens in the input sequence, with the model learning to predict these masked tokens based on surrounding context. This method benefits from bidirectional context, enabling consideration of both past and future tokens during predictions.

### 4.1.1 Causal Language Modeling

Borisov et al. (2023b) proposes the first CLM-based table generative method, GReaT[11] (Generation of Realistic Tabular data) to generate synthetic samples with original tabular data characteristics. The GReaT data pipeline involves a textual encoding step transforming tabular data into meaningful text using the sentences serialization methods as shown in Table 1, followed by fine-tuning a GPT-2 or GPT-2 distill model. Additionally, a feature order permutation step precedes the use of obtained sentences for LLM fine-tuning. REaLTabFormer (Solatorio & Dupriez, 2023) extends GReaT by generating synthetic non-relational and relational tabular data. It uses GReaT (an autoregressive GPT-2 model) to generate a parent table and a sequence-to-sequence model conditioned on the parent table for the relational dataset. The model implements target masking to prevent data copying and introduces statistical methods to detect overfitting. It demonstrates superior performance in capturing relational structures and achieves state-of-the-art results in predictive tasks without needing fine-tuning. Following the similar paradigm, Zhang et al.

---

[11]The code is in `https://github.com/kathrinse/be_great`

(2023e) proposed the TAPTAP[12] (Table Pretraining for Tabular Prediction) which incorporates several enhancements. The method involves continue pretraining the GPT2 on 450 Kaggle/UCI/OpenML tables, generating label columns using a machine learning model. Other improvements improvements include a revised numerical encoding scheme and the use of external models like gradient-boosted decision trees for pseudo-label generation. Their experimental findings demonstrate that by incorporating the additional table pre-training phase and employing machine learning models to generate labels, TAPTAP can generate superior quality training samples compared with GReaT. TabuLa (Zhao et al., 2023f), on the other hand, addresses the slow training of LLMs by using a randomly initialized model as the starting point; the method achieves continuous refinement through iterative fine-tuning on successive tabular data tasks [13]. It introduces a token sequence compression method and a middle padding strategy to simplify training data representation and enhance performance, achieving a significant reduction in training time while maintaining or improving synthetic data quality.

In contrast to above methods that fine-tune an LLM on the corresponding table samples, Curated LLM (CLLM) (Seedat et al., 2023) utilizes the rich world knowledge from GPT4 to augment and enhance training data in scenarios with limited data, without the need for fine-tuning. CLLM is a framework that leverages learning dynamics and two novel curation metrics, namely confidence and uncertainty. These metrics help to filter out undesirable generated samples during the training process of a classifier, aiming to produce high-quality synthetic data. Specifically, both metrics are calculated for each sample, utilizing the classifier trained on these samples. Additionally, CLLM distinguishes itself by not requiring any fine-tuning of LLMs.

**The generation process.** After fine-tuning the model or using a standard LLM, there are three primary preconditioning methods (Borisov et al., 2023b) for designing prompts to generate new data samples for CLM-based methods, as depicted in Figure 5: 1) feature name preconditioning: This method involves providing only a feature's name, generating samples across the entire joint data distribution. 2) One name-value pair preconditioning: Here, when a single feature name along with its value is supplied, the LLM will generate a complete sample. This method produces samples from the conditional distribution. Sampling one data point from a single feature distribution is generally feasible and then use name-value pair Preconditioning to generate the rest of the features. 3) Multiple Name-Value Pair Preconditioning: This involves providing multiple name-value pairs for arbitrary conditioning. The model then efficiently samples from the distribution of the remaining features. After that, we use cell value extraction methods, such as standard pattern-matching algorithms and regular expressions, to transform the generated pre-defined serialized text data into a tabular format.

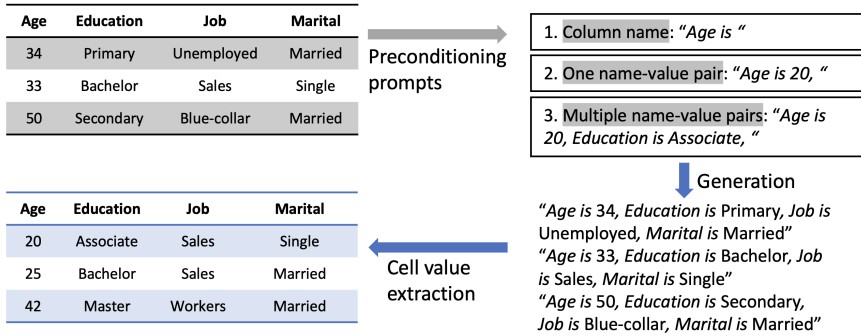

Figure 5: The data generation process for causual LMs

### 4.1.2 Masked Language Modeling

The MLM structure is suitable for generating tabular data due to its ability to capture bidirectional patterns between columns. Besides, prompting a tabular generator doesn't follow a sequential format. MLM's masking procedure enables arbitrary prompts during generation, making the generation process more efficient.

---

[12]The code is in `https://github.com/ZhangTP1996/TapTap`
[13]The code is in `https://github.com/zhao-zilong/Tabula`

Moreover, MLM can easily address the common challenge of missing data in tabular datasets by learning from missing values through a masking probability setting of 1, streamlining the generation process without requiring separate data imputation steps.

TabMT (Gulati & Roysdon, 2023) employs a masked transformer-based architecture. The design allows efficient handling of various data types and supports missing data imputation. It leverages a masking mechanism to enhance privacy and data utility, ensuring a balance between data realism and privacy preservation. TabMT's architecture is scalable, making it suitable for diverse datasets and demonstrating improved performance in synthetic data generation tasks.

**The generation process.** To minimize bias from a fixed order of column names, TabMT randomly selects column names without replacement during the generation process and subsequently samples the column values based on the predicted column distribution. TabMT initially sets the masking probability for all column values to 1 and then predicts each value gradually, as illustrated in Figure 6.

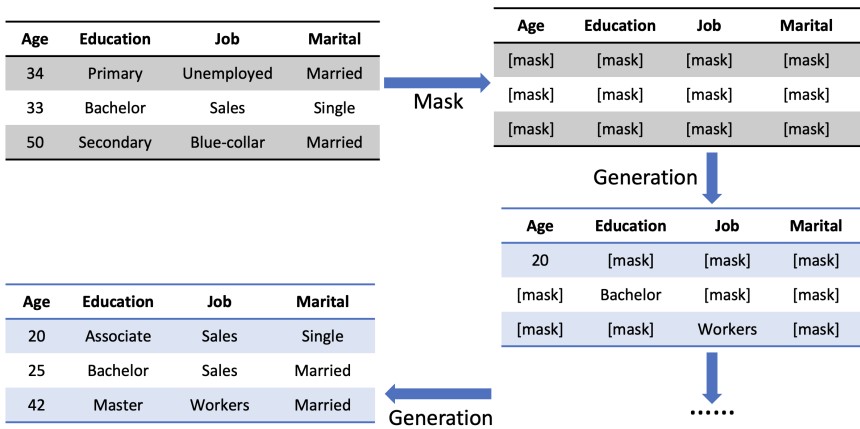

Figure 6: The data generation process for masked LMs

## 4.2 Evaluation

As outlined in Zhang et al. (2023c), the evaluation of synthetic data quality can be approached from four different dimensions: 1) **Low-order statistics** – *column-wise density* and *pair-wise column correlation*, estimating individual column density and the relational dynamics between pairs of columns, 2) **High-order metrics** – the calculation of $\alpha$-*precision* and $\beta$-*recall* scores that measure the overall fidelity and diversity of synthetic data, 3) **privacy preservation** – DCR score, representing the median Distance to the Closest Record (DCR), to evaluate the privacy level of the original data, (Note: the similarity-based DCR score provides an average metric for the system but does not offer information about individual privacy guarantees (Ganev & De Cristofaro, 2023)) and 4) Performance on **downstream tasks** – like *machine learning efficiency* (MLE) and *missing value imputation*. MLE is to compare the testing accuracy on real data when trained on synthetic ones. Additionally, the quality of data generation can be assessed through its performance on missing value imputation, that is, guessing the missing value(s) of a tuple, when the rest of the attributes are given.

## 5 LLMs for table understanding

In this section, we cover datasets, trends and methods explored by researchers for question answering (QA), fact verification (FV) and table reasoning tasks. There are many papers working on database manipulation, management and integration (Lobo et al., 2023; Fernandez et al., 2023; Narayan et al., 2022; Zhang et al., 2023b), which also include instructions and tabular inputs to LLMs. However, they are not typically referred to as a QA task, and they will not be covered by this paper.

## 5.1 Dataset

Table 7 outlines some of the popular datasets and benchmark in the literature working on tabular QA tasks. Other relevant but less commonly cited datasets are mentioned below.

| Dataset | # Tables | Task Type | Input | Output | Data Source | Evaluation Metric & Best Scores Reported |
|---|---|---|---|---|---|---|
| FetaQA (Nan et al., 2022) | 10330 | QA | Table Question | Answer | Wikipedia | BLEU: 39.05 (Zhang et al., 2023g), 35.12 (Sarkar & Lausen, 2023), 30.92 (Ye et al., 2023b), 27.02 (Chen, 2023), |
| WikiTableQuestion (Pasupat & Liang, 2015a) | 2108 | QA | Table Question | Answer | Wikipedia | Execution Accuracy: 73.65 (Liu et al., 2023e), 67.31 (Wang et al., 2024), 65.90 (Ye et al., 2023b), 65.90 (Jiang et al., 2023), 62.45 (Sarkar & Lausen, 2023), 48.80 (Chen, 2023), 35.01 (Zhang et al., 2023g) |
| HybridQA (Chen et al., 2020b) | 13000 | QA | Table Question | Answer | Wikipedia | Exact Match Accuracy: 39.38 (Zhang et al., 2023g), 25.14 (Sui et al., 2023c) |
| SQA (Iyyer et al., 2017) | 982 | QA | Table Question | Answer | WikiTable-Question | Exact Match Accuracy: 71.23 (Sarkar & Lausen, 2023), 33.45 (Sui et al., 2023c) |
| HiTAB (Cheng et al., 2022) | 3597 | QA/ NLG | Question, Table | Answer | Statistical Report and Wikipedia | Execution Accuracy: 64.71 (Zhang et al., 2023g), 50.00 (Zhao et al., 2023a) |
| ToTTo (Parikh et al., 2020a) | 120000 | NLG | Table | Sentence | Wikipedia | BLEU: 53.21 (Sui et al., 2023c), 20.77 (Zhang et al., 2023g) |
| FEVEROUS Aly et al. (2021) | 28800 | Classification | Claim, Table | Label | Wikipedia | Exact Match Accuracy: 77.22 (Chen, 2023), 73.77 (Zhang et al., 2023g), 66.51 (Sui et al., 2023c) |
| TabFact (Chen et al., 2020a) | 16573 | NLI | Table, Statement | Label | Wikipedia | Exact Match Accuracy: 93.00 (Ye et al., 2023b), 90.71 (Sarkar & Lausen, 2023), 87.60 (Jiang et al., 2023), 82.55 (Zhang et al., 2023g), 78.80 (Chen, 2023), 62.67 (Sui et al., 2023c) |
| Spider (Yu et al., 2018b) | 1020 | Text2-SQL | Table, Question | SQL | Human annotation | Execution Accuracy: 87.60 (Li et al., 2024), 86.60 (Gao et al., 2024), 85.30 (Pourreza & Rafiei, 2023), 82.30 (Dong et al., 2023), 79.90 (Li et al., 2023a), 78.00 (Rai et al., 2023), 77.80 (Jiang et al., 2023), 77.60 (Li et al., 2023b), 76.80 (Chang & Fosler-Lussier, 2023) |
| WikiSQL (Zhong et al., 2017b) | 24241 | Text2-SQL | Table, Question | SQL, Answer | Human Annotated | Execution Accuracy: 90.86 (Sarkar & Lausen, 2023), 65.60 (Jiang et al., 2023), 50.48 (Zhang et al., 2023g) |
| BIRD (Li et al., 2023c) | 694 | Text2-SQL | Table, Question, Evidence | SQL, Answer | Human Annotated | Execution Accuracy and Valid Efficiency Score. See https://bird-bench.github.io/ |

Table 7: Overview of popular QA/ reasoning datasets and related work for LLMs that worked on tabular data. Only datasets that have been used by more than one relevant methods are included in this table.

**Table QA** For table QA datasets, FetaQA (Nan et al., 2022), WikiTableQuestion (Pasupat & Liang, 2015a), HybridQA (Chen et al., 2020b) and SQA (Iyyer et al., 2017) are popular options. Unlike WikiTableQuestions, which focuses on evaluating a QA system's ability to understand queries and retrieve short-form answers from tabular data, FeTaQA introduces elements that require deeper reasoning and integration of information. This includes generating free-form text answers that involve the retrieval, inference, and integration of multiple discontinuous facts from structured knowledge sources like tables. This requires the model to generate long, informative, and free-form answers. NQ-TABLES Herzig et al. (2021) is larger than the previously mentioned

table. Its advantage lies in its emphasis on open-domain questions, which can be answered using structured table data.[14].

**Table and Conversation QA** HybriDialogue (Nakamura et al., 2022) includes conversations grounded on both Wikipedia text and tables. This addresses a significant challenge in current dialogue systems: conversing on topics with information distributed across different modalities, specifically text and tables. [15]

**Table Classification** FEVEROUS (Aly et al., 2021) focuses on both unstructured text and structured tables for fact extraction and verification tasks. Dresden Web Tables (Eberius et al., 2015)is useful for tasks requiring the classification of web table layouts, particularly useful in data extraction and web content analysis where table structures are crucial. The dataset is in footnote. [16]

**Text2SQL** Spider (Yu et al., 2018b), Magellan (Das et al., 2015) or WikiSQL (Zhong et al., 2017b), and BIRD (Li et al., 2023c) are suitable for training and evaluating models that generate SQL commands. Both Spider and WikiSQL have been benchmarked by many existing methods, some shown in Table 7. Compared to Spider, WikiSQL is much larger in size.[17]. The BIRD (BIg Bench for LaRge-scale Database Grounded Text-to-SQL Evaluation) benchmark contains large tables and complex questions, and it been widely used by the community.

**Table NLG** ToTTo (Parikh et al., 2020a) aims to create natural yet faithful descriptions to the source table. It is rich in size and can be used to benchmark table conditional text generation task. HiTAB (Cheng et al., 2022) allows for more standardized and comparable evaluation across different NLG models and tasks, potentially leading to more reliable and consistent benchmarking in the field. The dataset is in footnote. [18].

**Table NLI** InfoTabs (Gupta et al., 2020) uses Wikipedia info-boxes and is designed to facilitate understanding of semi-structured tabulated text, which involves comprehending both text fragments and their implicit relationships. InfoTabs is particularly useful for studying complex, multi-faceted reasoning over semi-structured, multi-domain, and heterogeneous data. Meanwhile, TabFact (Chen et al., 2020a) consists of human-annotated natural language statements about Wikipedia tables. It requires linguistic reasoning and symbolic reasoning to get right answer. The dataset is in footnote. [19].

**Domain-Specific** Some datasets and task focus on domain-specific applications. AIT-QA (Katsis et al., 2022) worked on airline industry specific table question answer. It highlights the unique challenges posed by domain-specific tables, such as complex layouts, hierarchical headers, and specialized terminology. For finance related table question answer, TAT-QA Zhu et al. (2021a) assesses numerical reasoning, involving operations like addition, subtraction, and comparison. SciGen (Moosavi et al., 2021) focuses on assessing the arithmetic reasoning capabilities of generation models on complex input structures, such as tables from scientific articles. TranX (Yin & Neubig, 2018) investigates abstract syntax description language for the target representations, enabling high accuracy and generalizability across different types of meaning representations.[20].

**Pretraining** For pretraining on large datasets for table understanding, we recommend to use TaBERT (Yin et al., 2020b) and TAPAS (Herzig et al., 2020). Dataset in Tapas has 6.2 million tables and is useful for

---

[14]Official sites for Table QA datasets: FetaQA (https://github.com/Yale-LILY/FeTaQA), WikiTableQuestions (https://ppasupat.github.io/WikiTableQuestions/), HybridQA (https://github.com/wenhuchen/HybridQA), SQA (https://www.microsoft.com/en-us/download/details.aspx?id=54253), and NQ-Tables (https://github.com/google-research-datasets/natural-questions).

[15]Official site for HybriDialogue: https://github.com/entitize/HybridDialogue

[16]Official site for FEVEROUS: https://fever.ai/dataset/feverous.html. Official site for Dresden Web Tables: https://ppasupat.github.io/WikiTableQuestions/.

[17]Official site for Spider: https://drive.usercontent.google.com/download?id=1iRDVHLr4mX2wQKSgA9J8Pire73Jahh0m&export=download&authuser=0. Official site for WikiSQL: https://github.com/salesforce/WikiSQL.

[18]Official site for ToTTo: https://github.com/google-research-datasets/ToTTo. Official site for HiTAB: https://github.com/microsoft/HiTab

[19]Official site for InfoTabs: https://infotabs.github.io/. Official site for TabFact: https://tabfact.github.io/

[20]Official sites for the domain-specific datasets: AIT-QA (https://github.com/IBM/AITQA), TAT-QA (https://github.com/NExTplusplus/TAT-QA), SciGen (https://github.com/UKPLab/SciGen) and TranX (https://github.com/pcyin/tranX)

semantic parsing. TAPAS has 26 million tables and their associated english contexts. It can help model gain better understanding in both textual and table. The dataset is in footnote. [21].

## 5.2 General ability of LLMs in QA

| Paper | Task | Models Explored |
|---|---|---|
| DOCMATH-EVAL (Zhao et al., 2023d) | NumQA | GPT4, GPT3.5, WizardLM, Llama-2 7, 13, 70B, CodeLlama 34B, Baichuan, Qwen, WizardMath, Vicuna, Mistral, etc. |
| Akhtar et al. (2023) | NumQA | TAPAS, DeBERTa, TAPEX, NT5, LUNA, PASTA, ReasTAP, FlanT5, GPT3.5, PaLM |
| TableGPT (Gong et al., 2020) | NumQA | GPT2 |
| DATER (Ye et al., 2023b) | QA | GPT3 Codex |
| PACIFIC (Deng et al., 2022b) | QA | T5, CodeT5 |
| Chen (2023) | QA | GPT3 |
| cTBLS (Sundar & Heck, 2023) | QA | Custom: Dense Table Retrieval based on RoBERTa + Coarse State Tracking + Response based on GPT3.5 |
| GPT4Table (Sui et al., 2023b) | QA | GPT-3.5, GPT-4 |
| Zhao et al. (2023a) | QA | GPT-3.5 |
| Liu et al. (2023e) | QA | GPT3.5 |
| TableGPT (Zha et al., 2023) | QA | Phoenix-7B |
| TAP4LLM (Sui et al., 2023c) | QA | Instruct GPT3.5, GPT4 |
| UniTabPT (Sarkar & Lausen, 2023) | QA | T5 |
| Yu et al. (2023) | Multi-modal QA | Custom: Retrieval trained on contrastive loss, Rank by softmax, Generation built on T5 |
| TableLlama (Zhang et al., 2023g) | QA | Custom: TableLlama |
| DIVKNOWQA (Zhao et al. (2023c) | QA | GPT3.5, DSP, ReAct |
| Jiang et al. (2023) | QA | GPT3.5, ChatGPT3.5 |
| Liu et al. (2023c) | QA & Text2SQL | Vicuna, GPT4 |
| OpenTab (Kong et al., 2024) | QA & Text2SQL | GPT-3.5-Turbo, Falcon-180B |
| Gao et al. (2024) | Text2SQL | GPT4 |
| Pourreza & Rafiei (2023) | Text2SQL | GPT4 |
| Huang et al. (2023b) | Text2SQL | GPT4 |
| Dong et al. (2023) | Text2SQL | ChatGPT3.5 |
| Chang & Fosler-Lussier (2023) | Text2SQL | GPT3 Codex, ChatGPT3.5 |
| Zhang et al. (2023d) | Text2SQL | LLaMA2 70b |
| Abraham et al. (2022) | Text2SQL | Custom: Table Selector + Known & Unknown Fields Extractor + AggFn Classifier |

Table 8: Overview of Papers and Models for LLMs for tabular QA tasks. We only include papers that work with models of >1B parameters. Models that are described as "Custom" indicates papers that finetuned specific portions of their pipeline for the task, whereas the other papers focus more on non-finetuning methods like prompt engineering. NumQA: Numerical QA.

Table 8 outlines papers that investigated the effectiveness of LLMs on QA and reasoning, and the models explored. The most popular LLM used today is GPT3.5 and GPT4. Although these GPT models were not specifically optimized for table-based tasks, many of these papers found them to be competent in performing complex table reasoning tasks, especially when combined with prompt engineering tricks like CoT. In this section, we summarize the general findings of LLMs in QA tasks and highlight models that have reported to work well.

**Numerical QA**  A niche QA task involves answering questions that require mathematical reasoning. An example query could be "*What is the average payment volume per transaction for American Express?*" Many real-world QA applications (E.g. working with financial documents, annual reports, etc.) involve such

---

[21]The dataset for TaBERT is in `https://github.com/facebookresearch/TaBERT`. The dataset for TAPAS is in `https://github.com/google-research/tapas`

mathematical reasoning tasks. So far, Akhtar et al. (2023) conclude that LLMs like FlanT5 and GPT3.5 perform better than other models on various numerical reasoning tasks. On the DOCMATH-EVAL (Zhao et al., 2023d) dataset, GPT-4 with CoT significantly outperforms other LLMs, while open-source LLMs (LLaMa-2, Vicuna, Mistral, Starcoder, MPT, Qwen, AquilaChat2, etc.) lag behind.

**Text2SQL** Liu et al. (2023c) designed a question matcher that identifies three keyword types: 1) column name-related terms, 2) restriction-related phrases (e.g. "top ten"), and 3) algorithm or module keywords. Once these keywords are identified, the module begins to merge the specific restrictions associated with each column into a unified combination, which is then matched with an SQL algorithm or module indicated by the third type of keyword. Zhang et al. (2023d) opted for a more straightforward approach of tasking LLaMa-2 to generate an SQL statement based on a question and table schema. Sun et al. (2023b) finetuned PaLM-2 on the Text2SQL task, achieving considerable performance on Spider. OpenTab (Kong et al., 2024) developed an LLM-based framework for the open-domain table QA tasks, incorporating a SQL generation Coder module. The top scoring models for the Spider today are Dong et al. (2023); Gao et al. (2024); Pourreza & Rafiei (2023), all building off OpenAI's GPT models. SQL generation is popular in the industry, with many open-source fine-tuned models available[22].

**Impact of model size on performance** Chen (2023) found that size does matter: On WebTableQuestions, when comparing the 6.7B vs. 175B GPT-3 model, the smaller model achieved only half the scores of the larger one. On TabFact, they found that smaller models ($<=$6.7B) obtained almost random accuracy.

**Finetuning or No finetuning?** There are some larger models that fine-tune on various tabular tasks, some including QA and FV tasks, mentioned in Section 2.1 under embeddings-based serialization. Li et al. (2023d) found that fine-tuning always helps to improve performance across various tabular tasks. In zero-shot settings, the improvement is the most significant. For Ye et al. (2023b), they obtained higher scores on TabFact when using their framework with the PASTA (Gu et al., 2022) model (score 93.00%) as compared to the GPT-3 Codex (code-davinci-002) (scored 85.60%). PASTA was pre-trained on a synthesized corpus of 1.2 million items from WikiTables for six types of sentence–table cloze tasks. This suggests there remains some benefit in using LMs fine-tuned on tabular tasks.

However, compared to methodologies working on Prediction and Generation tasks, fine-tuning is not as common. This might be due to the general ability of LLMs (E.g. GPT3.5, GPT4) to perform QA tasks off-the-shelf. For SQL generation on Spider, DIN-SQL (Pourreza & Rafiei, 2023) and DAIL-SQL (Gao et al., 2024) are inference-based techniques using GPT4, and surpassed previous fine-tuned smaller models. Interestingly, in the paper by Gao et al. (2024), the authors fine-tuned a Llama 2 13B model on the Text2SQL tasks. However, this model did not beat the GPT4 model that was not fine-tuned. Instead, many papers working on using LLMs for table understanding tasks focus on tweaking aspects across serialization, prompt engineering, search and retrieval, and end-to-end pipelines (user interfaces), which we describe further in the next section.

### 5.3 Key components in Tabular QA

In the simplest QA architecture, an LLM takes in an input prompt (query and serialized table)[23], and returns an answer. In more involved architectures, the system might be connected to external databases or programs. Most of the times, the knowledge base might not fit in the context length or memory of the LLM. Therefore, unique challenges to tabular QA for LLMs include: query intent disambiguation, search and retrieval, output types and format, and multi-turn settings where iterative calls between programs are needed. We describe these components further in this section.

#### 5.3.1 Query intent disambiguation

Zha et al. (2023) introduced the concept of Chain-of-command (CoC), that translates user inputs into a sequence of intermediate command operations. For example, an LLM needs to first check if the task

---

[22]https://huggingface.co/NumbersStation
[23]For the scope of our paper, we do not consider images, videos and audio inputs.

requires retrieval, mathematical reasoning, table manipulations, and/or the questions cannot be answered if the instructions are too vague. They constructed a dataset of command chain instructions to fine-tune LLMs to generate these commands. Deng et al. (2022b) proposed the QA task be split into three subtasks: Clarification Need Prediction (CNP) to determine whether to ask a question for clarifying the uncertainty; Clarification Question Generation (CQG) to generate a clarification question as the response, if CNP detects the need for clarification; and Conversational Question Answering (CQA) to directly produce the answer as the response if it is not required for clarification. They trained a UniPCQA model which unifies all subtasks in QA through multi-task learning.

### 5.3.2   Search and retrieval

The ability to accurately search and retrieve information from specific positions within structured data is crucial for LLMs. There are two types of search and retrieval use-cases: (1) to find the information (table, column, row, cell) relevant to the question, and (2) to obtain additional information and examples.

**For main table**   Zhao et al. (2023d) observed that better performance of a retriever module (that returns the top-n most relevant documents) consistently enhances the final accuracy of LLMs in numerical QA. Sui et al. (2023c) explored multiple table sampling methods (of rows and columns) and table packing (based on a token-limit parameter). The best technique was the query-based sampling, which retrieves rows with the highest semantic similarity to the question, surpassing methods involving no sampling, or clustering, random, even sampling, or content snapshots. Dong et al. (2023) used ChatGPT to rank tables based on their relevance to the question using SC: they generate ten sets of retrieval results, each set containing the top four tables, then selecting the set that appears most frequently among the ten sets. To further filter the columns, all columns are ranked by relevance to the question by specifying that ChatGPT match the column names against with the question words or the foreign key should be placed ahead to assist in more accurate recall results. Similarly, SC method is used. cTBLS Sundar & Heck (2023) designed a three-step architecture to retrieve and generate dialogue responses grounded on retrieved tabular information. In the first step, a dual-encoder-based Dense Table Retrieval (DTR) model, initialized from RoBERTa Liu et al. (2019), identifies the most relevant table for the query. In the second step, a Coarse System State Tracking system, trained using triplet loss, is used to rank cells. Finally, GPT-3.5 is prompted to generate a natural language response to a follow-up query conditioned on cells of the table ranked by their relevance to the query as obtained from the coarse state tracker. The prompt includes the dialogue history, ranked knowledge sources, and the query to be answered. Their method produced more coherent responses than previous methods, suggesting that improvements in table retrieval, knowledge retrieval, and response generation lead to better downstream performance. Zhao et al. (2023d) used OpenAI's Ada Embedding4 and Contriever (Izacard et al., 2022) as the dense retriever along with BM25 (Robertson et al., 1995) as the sparse retriever. These retrievers help to extract the top-n most related textual and tabular evidence from the source document, which were then provided as the input context to answer the question.

**For additional information**   Some papers explore techniques to curate samples for in-context learning. Gao et al. (2024) explored the a few methods: (1) random: randomly selecting $k$ examples; (2) question similarity selection: choosing $k$ examples based on semantic similarity with question $Q$, based on a predefined distance metric (E.g. Euclidean or negative cosine similarity) of the question and example embedding, and kNN algorithm to select $k$ closest examples from $Q$; (3) masked question similarity selection: similar to (2), but beforehand masking domain-specific information (the table names, column names and values) in the question; (4) query similarity selection: select $k$ examples similar to target SQL query $s*$, which relies on another model to generate SQL query $s'$ based on the target question and database, and so $s'$ is an approximation for $s*$. Output queries are encoded into binary discrete syntax vectors. Narayan et al. (2022) explored manually curated and random example selection.

### 5.3.3   Multi-turn tasks

Some papers design pipelines that call LLMs iteratively. We categorize the use-cases for doing so into three buckets: (1) to decompose a challenging task into manageable sub-tasks, (2) to update the model outputs based on new user inputs, and (3) to work-around specific constraints or to resolve errors.

**Intermediate, sub-tasks** This section overlaps with concepts around CoT and SC discussed earlier in Section 2.3. In a nutshell, since the reasoning task might be complex, LLMs might require guidance to decompose the task into manageable sub-tasks. For example, to improve downstream tabular reasoning, Sui et al. (2023b) proposed a two-step self-augmented prompting approach: first using prompts to ask the LLM to generate additional knowledge (intermediate output) about the table, then incorporating the response into the second prompt to request the final answer for a downstream task. Ye et al. (2023b) also guided the LLM to decompose a huge table into a small table, and to convert a complex question into simpler sub-questions for text reasoning. Their strategy achieved significantly better results than competitive baselines for table-based reasoning, outperforms human performance for the first time on the TabFact dataset. For Liu et al. (2023e), in encouraging symbolic CoT reasoning pathways, they allowed the model to interact with a Python shell that could execute commands, process data, and scrutinize results, particularly within a pandas dataframe, limited to a maximum of five iterative steps.

**Dialogue-based applications** In various applications where the users are interacting with the LLMs, like in chatbots, the pipeline must allow for LLMs to be called iteratively. Some dialogue-based Text2SQL datasets to consider are the SParC (Yu et al., 2019b) and CoSQL (Yu et al., 2019a) datasets. For SParC, the authors designed subsequent follow-up questions based on Spider (Yu et al., 2018b).

**Working around constraints or error de-bugging** Zhao et al. (2023a) used multi-turn prompts to work around cases where the tables exceed the API input limit. In other cases, especially if the generated LLM output is code, an iterative process of feeding errors back to the LLM can help the LLM generate correct code. Zhang et al. (2023d) did so to improve SQL query generation.

### 5.3.4 Output evaluation and format

If the QA output is a number or category, F1 or Accuracy evaluation metrics are common. If evaluating open-ended responses, apart from using typical measures for like ROUGE and BLEU, some papers also hire annotators to evaluate the Informativeness, Coherence and Fluency of the LLM responses Zhang et al. (2023h). When connected to programs like Python, Power BI, etc, LLMs' outputs are not limited to text and code. For example, creating visualizations from text and table inputs are a popular task too Zhang et al. (2023h); Zha et al. (2023).

## 6 Limitations and future directions

LLMs have been used in many tabular data applications, such as predictions, data synthesis, question answering and table understanding. Here we outline some practical limitations and considerations for future research.

**Numerical representation** It was revealed that LLM in house embedding is not suitable for representing intrinsic relations in numerical features (Gruver et al., 2023), and thus a careful embedding is needed. Tokenization significantly impacts pattern formation and operations in language models. Traditional methods like Byte Pair Encoding (BPE) used in GPT-3 often split numbers into non-aligned tokens (e.g., 42235630 into [422, 35, 630]), complicating arithmetic. Newer models like LLaMA tokenize each digit separately. Both approaches make LLM difficult to understand the whole number. Also, based on Spathis & Kawsar (2023), the tokenization of integers lacks a coherent decimal representation, leading to a fragmented approach where even basic mathematical operations require memorization rather than algorithmic processing. The development of new tokenizers, like those used in LLaMA (Touvron et al., 2023b), which outperformed GPT-4 in arithmetic tasks, involves rethinking tokenizer design to handle mixed textual and numerical data more effectively, such as by splitting each digit into individual tokens for consistent number tokenization (Gruver et al., 2023). This method has shown promise in improving the understanding of symbolic and numerical data. However, it hugely increases the dimension of the input, which makes the method not practical for large datasets and many features. For future direction, it is worth to explore new tokenizer that can better represent numerical token while not increase the dimension of the input.

**Categorical representation** Tabular dataset very often contains an excessive number of categorical columns, which can lead to serialized input strings surpassing the context limit of the language model and increased cost. This is problematic as it results in parts of the data being pruned, thereby negatively impacting the model's performance. Additionally, there are issues with poorly represented categorical feature, such as nonsensical characters, which the model struggles to process and understand effectively. Another concern is inadequate or ambiguous metadata, characterized by unclear or meaningless column names and metadata, leading to confusion in the model's interpretation of inputs. Better categorical features encoding is needed to solve these problems. Traditional machine learning methods such as lightGBM require expanding dimension for categorical features (Borisov et al., 2022b) and can lead to bias categorical representation (Prokhorenkova et al., 2019). Thus, good categorical features encoding could add competitive advantage for LLM based method compared to traditional machine learning methods.

**Standard benchmark** LLMs for tabular data could greatly benefit from standardized benchmark datasets to enable fair and transparent comparisons between models. In this survey, we strive to summarize commonly used datasets/metrics and provide recommendations for dataset selection to researchers and practitioners. However, for the same dataset, the same method, and the same task, different papers report different performance. For prediction task, the performance of TabLLM (Hegselmann et al., 2023) in Blood dataset (Kadra et al., 2021) is 0.70 in GTL (Zhang et al., 2023a) (see table 2 in that paper) and 0.66 in UniPredict (Wang et al., 2023a) (see table 4 in that paper). This discrepancy in benchmark performance makes it impossible to come up with a classification performance benchmark against all methods. Therefore, there is a pressing need for more standardized and unified benchmark exercise to bridge this gap effectively.

**Tabular-specific challenges** The current exploration of LLMs on tabular data remains primarily surface-level, lacking in-depth analysis tailored to the unique characteristics of tabular datasets. For example, there is a paucity of understanding regarding how LLMs handle class imbalanced datasets. Given that LLMs come with prior knowledge, it is reasonable to hypothesize about the synergistic or antagonistic effects between training and inference data, potentially leading to unforeseen behaviors in such scenarios (Jung & van der Plas, 2024). Another unexplored aspects relates to the order invariant nature of tabular data. while language models are inherently order-variant, with word order significantly impacting predictions and contextual understanding, little is unknown how LLM performance varies when dealing with tabular data where orders of the features and records are invariant. Future research should prioritize an in-depth investigation into tabular-specific behaviors of LLMs to enhance their performance on tasks related to tabular data.

**Bias and fairness** In existed tabular prediction and table understanding methods, LLMs tend to inherit social biases from their training data, which significantly impact their fairness metric. For example, Liu et al. (2023f) uses GPT3.5 and do few-shot learning to evaluate the fairness of tabular prediction on in context learning. For LLMs based tabular prediction method, the research concludes that the fairness metric gap between different subgroups is larger than that in traditional machine learning model. Additionally, the research further reveals that flipping the labels of the in-context examples significantly narrows the gap in fairness metrics across different subgroups, but comes at the expected cost of a reduction in predictive performance. Other research shows that the inherent bias of LLM is hard to mitigate through prompt (Hegselmann et al., 2023). Thus, it is worth to explore through other bias mitigation methods such as pre-processing (Shah et al., 2020) or optimization (Bassi et al., 2024).

**Hallucination** LLMs sometimes produce content that is inconsistent with the real-world facts or the user inputs (Huang et al., 2023a), which raises concerns over the reliability and usefulness of LLMs in the real-world applications. For tabular prediction, especially when working with patient records and medical data, hallucinations have critical consequences. Akhtar et al. (2023) found that hallucination led to performance drops in reasoning for LLMs. To address these issues in tabular prediction, Wang et al. (2023c) incorporated an audit module that utilizes LLMs to perform self-check and self-correction. They generated pseudo-labels, then used a data audit module which filters the data based on data Shapley scores, leading to a smaller but cleaner dataset. Secondly, they removed any cells with False values, which removes the chances of the LLMs making a false inference on these invalid values. Finally, they performed a sanity check via LLM's reflection: querying the LLM with the input template *"What is the {column}? {x}"* to check if the answer matches the original values. If the answers do not match, the descriptions are corrected by re-prompting the LLM. However, this method requires iterative efforts and is hard to deploy in real world application.

An interesting future direction would be to explore efficient and practical way to deal with hallucination in LLM based method for tabular data.

**Model interpretability** Like many deep learning algorithms, LLMs suffer from a lack of interpretability. For LLM based method in tabular data, only a few systems expose a justification of their model output, such as TabLLM (Hegselmann et al., 2023). One direction is to use the Shapley values to derive interpretations. Shapley values have been used to evaluate the prompt for LLMs (Liu et al., 2023a). It could also be useful to understand how each feature influence the result. For instance, in prediction for diseases, providing explanation is crucial. In this case, an explanation based on Shapley values would list the most important features that led to the final decision. However, the performance of Shapley or other explanation methods on tabular prediction and table understanding remains unexplored. Future research is needed to explore the existed explanation mechanisms for LLM based tabular prediction and table understanding and develop more suited explanation methods.

**Ease of use** Existed LLM based tools such as ChatGPT and models on huggingface are easy to inference. Currently, most relevant tabular based LLM models require fine-tuning or data serialization, which could make these models hard to access. Pretrained models, such as Wang et al. (2023c;a) which integrate data consolidation, enrichment, and refinement, have the potential to streamline user experience. These methods still require extensive preprocessing, which makes it hard to inference. The development of a unified pipeline that incorporates these models, along with auto data prepossessing and serialization to established platforms such as Hugging Face, warrants further exploration.

**Fine-tuning strategy design** Designing appropriate tasks and learning strategies for LLMs is extensively explored. LLMs demonstrate emergent abilities such as in-context learning, instruction following, and step-by-step reasoning. However, these capabilities may not be fully evident in certain tabular data prediction and table understanding tasks, depending on the model used. LLMs are sensitive to various serialization and prompt engineering methods (Hegselmann et al., 2023), which is the primary way to adapt LLM to unseen tasks. As a future direction, researchers and practitioners need to carefully design tasks and learning strategies tailored to tabular data in order to achieve an optimal performance.

**Model grafting** The performance of LLM for tabular data modeling could be improved through model grafting. Model grafting involves mapping non-text data into the same token embedding space as text using specialized encoders, as exemplified by the HeLM model (Belyaeva et al., 2023), which integrates spirogram sequences and demographic data with text tokens. This approach is efficient and allows integration with high-performing models from various domains but adds complexity due to its non-end-to-end training nature and results in communication between components that is not human-readable. This approach could be adapted to LLM for tabular data to improve the encoding of non-text data such as categorical and numerical feature.

## 7 Conclusion

This survey represents the first large-scale investigation into the utilization of LLMs for modeling heterogeneous tabular data across various tasks, including prediction, data synthesis, question answering and table understanding. We delve into the essential steps required for tabular data to be ingested by LLM, covering serialization, table manipulation, and prompt engineering. Additionally, we systematically compare datasets, methodologies, metrics and models for each task, emphasizing the principal challenges and recent advancements in understanding, inferring, and generating tabular data. We provide recommendations for dataset and model selection tailored to specific tasks, aimed at aiding both ML researchers and practitioners in selecting appropriate solutions for tabular data modeling using different LLMs. Moreover, we examine the limitations of current approaches, such as susceptibility to hallucination, fairness concerns, data preprocessing intricacies, and result interpretability challenges. In light of these limitations, we discuss future directions that warrant further exploration in future research endeavors.

With the rapid development of LLMs and their impressive emergent capabilities, there is a growing demand for new ideas and research to explore their potential in modeling structured data for a variety of tasks. Through this comprehensive review, we hope it can provide interested readers with pertinent references and insightful perspectives, empowering them with the necessary tools and knowledge to effectively navigate and

address the prevailing challenges in the field. We acknowledge that the references provided are by no means an exhaustive list, but rather represent topics that have been widely discussed. We apologize to any related papers that have not been included.

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
