# OpenReview forum: "Large Language Models (LLMs) on Tabular Data: Prediction, Generation, and Understanding - A Survey"
_TMLR — Accepted by TMLR_

### Review · Reviewer_mkwJ · 2024-04-18

**Summary Of Contributions:**

This paper investigates the ability of large language models for tabular data. It categorizes the mainstream tasks into three categories: prediction, generation, and understanding. Then, it summarizes current studies, techniques, benchmarks, and metrics on each task. The authors also discussed the limitations of existing tabular data analysis driven by LLMs, including hallucination, and categorical/numerical representations.

**Audience:**

Yes

**Claims And Evidence:**

Yes

**Requested Changes:**

1. This paper requires major revision, especially improved writing.

2. Section 2 should be revised.

3. The authors should provide more in-depth discussion and their own opinions on the technologies and future directions.

**Strengths And Weaknesses:**

Pros:

1.	Tabular data learning is an important research problem and this survey may inspire future research in this area.

2.	It summarizes most current papers on tabular data learning papers driven by LLMs.

Cons:

1.	The organization and intuition of Section 2 is terrible.

a)	In Section 2.1 (Serialization), the most papers are traditional techniques in tabular learning.

b)	Section 2.2: What is table manipulation? Please give a clear definition.

c)	Section 2.3: Prompt engineering is not an application of LLMs, but a part of LLM.

d)	It is unclear what the author intends to express in this section.

e)	This section mentioned (Sui et. al. 2023) many times and it seems most conclusions are from (Sui et. al. 2023).

2.	The presentation is terrible. Many writing issues (only a few examples):

a)	Figure 3/4 is very blurry; was it copied from another paper?

b)	Page 13: what is ‘t few(?)’

c)	Page 19: missing brackets at ‘Das et. al.’

d)	Page 14: what does ‘we are left with 115 tabular datasets … we randomly sample ….’ mean? I found the authors wrongly copies the sentences from TabFMs. The authors should not blindly copy others' text in a review.

3.	In section 4.1, the author merely lists the methods and appears to simply describe the content of the original text without conducting a detailed analysis and discussion. This problem arises everywhere in the survey, making it lack of depth. For example, in section 3.3, the authors suddenly mentions pseudo-labels without clear contexts.

4.	For example, the limitations section also simply lists some common issues in the area of LLM research, e.g. hallucination and fine-tuning. I don’t find many future directions in this section.

---

> ### Author Response · Authors · 2024-05-10
> **Update the paper based on requested change and weakness**
>
> Thanks the reviewer for their thoughtful feedback! We edited the paper based on reviewer's feedback.
>
> 1. The organization and intuition of Section 2 is terrible.
> Answer: Regarding your feedback on Section 2, we have heeded your advice to improve the organization and intuition of the Section. Firstly, we included a flowchart to help readers visualize the common techniques of using LLMs for tabular data. Additionally, here are our responses to your respective comments:
>
> •	The papers under Section 2.1 cover very current publications from 2020 to 2023. Even if these techniques are traditional, we believe introducing these common serialization approaches in the literature is necessary to help readers understand the steps in using LLMs for tabular data better.
> •	Thank you for your advice. We have included a clear definition for table manipulation in the paper.
> •	We would like to clarify that we mentioned prompt engineering as a “key technique” of using LLMs for tabular data. We did not mention that prompt engineering is an application of LLMs.
> •	All in all, Section 2 introduces common techniques pre-emptively, facilitating smoother transitions into task-specific discussions in subsequent sections. We hope that this intention is expressed more clearly now with the changes made.
> •	We would like to clarify that there are three distinct papers published by Sui et al. in 2023, each researching different aspects of tabular data and LLMs. Therefore, it seemed like conclusions were drawn from Sui et al. often. Nevertheless, we have included a few more citations in Section 2 to reduce the impression that our conclusions are predominantly from Sui et al.
>
> 2. The presentation is terrible. Many writing issues
> Answer: Thank you very much for pointing these out. We realize there was some overlap in language when introducing the data selection strategy of TabFMs. We have revised the language throughout and revised all the errors.
>
> 3. In section 4.1, the author merely lists the methods and appears to simply describe the content of the original text without conducting a detailed analysis and discussion.
>
> Answer: In the section on table generation, we enhance the description by adding the motivation behind the task and providing deeper insights into existing methods. In this section we survey methodologies that leverage LLMs for tabular data synthesis.
> Please see the section 4.1 for details.
>
> 4. The limitations section also simply lists some common issues in the area of LLM research, e.g. hallucination and fine-tuning. I don’t find many future directions in this section.
>
> Answer: All those limitations are for tabular data where we have grounded each limitation with our surveyed work. In the new version section 6, we explicitly explain those limitation in tabular data space. For numerical representation, the tokenization of integers lacks a coherent decimal representation and some model (GPT3) split numbers into non-aligned tokens. For bias and fairness and tabular data prediction, the fairness metric gap between different subgroups is larger than that in traditional machine learning model. For model interpretability, only a few systems expose a justification of their model output and we cannot find a standard way to interpret LLMs in tabular prediction and table understanding. For ease of use, LLM based tabular methods still require extensive preprocessing, which makes it hard to inference. For Fine-tuning strategy, existed strategy for LLM may not be fully evident in certain tabular data prediction and table understanding tasks, depending on the model used. For categorical representation, there are issues with poorly represented categorical feature, such as nonsensical characters, which the model struggles to process and understand effectively. For standard benchmark, for the same dataset, the same method, and the same task, different papers report different performance. For prediction task, the performance of TabLLM (Hegselmann et al., 2023) in Blood dataset (Kadra et al., 2021) is 0.70 in GTL (Zhang et al., 2023a) (see table 2 in that paper) and 0.66 in UniPredict (Wang et al., 2023a) (see table 4 in that paper). This discrepancy in benchmark performance makes it impossible to come up with a classification performance benchmark against all methods. We have also added a paragraph regarding ‘tabular-specific challenges’ and pointed out tabular features like class imbalance and order invariant is an important area for future research.
> To incorporate your suggestion and make future directions more visible, we have highlighted future directions grounded by surveyed work and practical experience in the last section of each paragraph.

---

### Review · Reviewer_irw6 · 2024-04-26

**Summary Of Contributions:**

This paper provides a survey on work that applies large language models to tasks and problems arising from tabular data. The paper covers a large body of work, organized into key techniques such as representing tables as text and fitting them into prompt length limits; about making predictions from tabular data; generating synthetic tabular data or imputing missing data; and question answering from tables. For each of these sections, the paper gives summaries of many of these other works and also provides some key information about them summarized into tables. The paper closes with discussion of some common limitations of the surveyed work and potential future directions.

**Audience:**

Yes

**Broader Impact Concerns:**

No broader impact concerns.

**Claims And Evidence:**

No

**Requested Changes:**

More important:
- Please include some quantitative comparisons of the works covered. For example, if multiple papers use the same dataset and try to make predictions using them, it should be possible to report their metrics in a summarized way. It would also be interesting to see how other variables affect the metrics (e.g. the amount of data and compute used).
- Section 2 is useful for covering work that is specifically about the questions explored there (e.g. about serializing tables), but those questions also need to be covered by work more about the downstream applications (because, for example to make predictions on financial data, with LLMs, you would also need to serialize the tables). It would be better if section 2 also served to summarize and compare all papers cited and discussed later in the paper where the issue of serializing tables (and other questions covered) are relevant.

Less important:
- Personally, I think it is uninteresting to list the specific LLMs evaluated in the cited papers, also considering - that they can take up a lot of space (for example, in Table 8). It should be enough to summarize briefly what kinds of models are compatible with their method or study. The paper can also summarize the model usage by counting how many other papers surveyed used each model, if readers are interested in a measure of popularity.
- Many citations are incorrectly formatted throughout the paper, for example using `\citet` when `\citep` would have been needed: for example, on page 19 in "**Table NLG** ToTTo Parikh et al. (2020a) aims to create"

**Strengths And Weaknesses:**

Strengths:
- The paper contains a huge number of references that can be useful for anyone looking for related work in this area, especially when their search is congruent with the taxonomy used in this paper.
- The paper, while quite lengthy, covers many different aspects of tabular data relating to LLMs, including both techniques and applications.
- The authors identify some common limitations discussed by several of the papers surveyed and discusses them.

Weaknesses:
- While the paper does provide some tables comparing and summarizing the surveyed works, much of the paper reads like short summaries of the other papers concatenated together. The authors could have made a deeper comparison between the works surveyed.
- There is very little summary or discussion of the empirical quality of the surveyed work, e.g. the classification performance achieved by the surveyed papers in the predictions section. It is hard to tell which methods might be more useful for not, and what tradeoffs they might have made (such as greater methodological complexity and additional computational power needed) to get there.
- The authors often frame their discussion of the work as recommendations, which seems inappropriate for a survey paper, especially because they are usually not grounded in quantitative reasoning even when that would be useful information to have. Ideally, the paper would present the required data that makes it easy for the reader to make such recommendations for themselves, taking into account their requirements and constraints (e.g. on amount of compute and task-specific data available).
- While some parts of the limitations and future directions section synthesize common issues faced by the surveyed work, other parts seem ungrounded in the surveyed work.

---

> ### Author Response · Authors · 2024-05-10
> **Update the paper based on requested change and weakness**
>
> Thanks the reviewer for their thoughtful feedback! We have edited the paper based on the reviewer's feedback.
>
> 1. While the paper does provide some tables comparing and summarizing the surveyed works, much of the paper reads like short summaries of the other papers concatenated together. The authors could have made a deeper comparison between the works surveyed.
>
>
> Answer: Thank you very much for the suggestions. In the new version, we have broken down the methods to different steps and discussed the connection and difference between different methods on each step.
>
> In the tabular prediction section, all prediction methods are broken down to different steps such as serialization, training, inference and target augmentation. We compared between examples on those steps and provide examples to illustrate the difference, as you can see in table 4 and 5.
>
> In the table generation section, we recategorize the methods into two typical classes, Causal Language Modeling (CLM)-powered methods and Masked Language Modeling (MLM)-powered methods. We illustrate the difference between different methods in figure 5 and 6.
>
> 2.  It is hard to tell which methods might be more useful for not, and what tradeoffs they might have made.
>
> Answer: For Tabular prediction section, we mentioned the resource requirement for each method which is based on complexity and computation power
>
> 3. The authors often frame their discussion of the work as recommendations, ... the paper would present the required data that makes it easy for the reader to make such recommendations for themselves, taking into account their requirements and constraints.
>
> Answer: To address your concern that we “often frame their discussion of the work as recommendations”, we have edited the wording and tone of Section 5 and 3. Specifically, we removed all comments(14 places) like “we recommend” to become descriptions of the literature. For example, “We recommend benchmarking FEVEROUS if the tasks involve fact verification using both unstructured text and structured tables.” à “FEVEROUS focuses on both unstructured text and structured tables for fact extraction and verification tasks.”
>
> 4. Please include some quantitative comparisons of the works covered.
>
> Answer: Thank you for your helpful advice to “include some quantitative comparisons of the works covered”. We have included the reported metrics for common QA and reasoning datasets in Section 5, under Table 7.  For tabular prediction, it is harder to include metrics in a summarized way. To be specific, for the same dataset, the same method, and the same task, different papers report different performance. For prediction task, the performance of TabLLM (Hegselmann et al., 2023) in Blood dataset (Kadra et al., 2021) is 0.70 in GTL (Zhang et al., 2023a) (see table 2 in that paper) and 0.66 in UniPredict (Wang et al., 2023a) (see table 4 in that paper). This discrepancy in benchmark performance makes it impossible to come up with a classification performance benchmark against all methods. We mention above to limitation section.
>
> 5. It would be better if section 2 also served to summarize and compare all papers cited and discussed later in the paper where the issue of serializing tables (and other questions covered) are relevant.
>
> Answer: We agree with your comment that Section 2 would be better if it "served to summarize and compare all papers cited and discussed later in the paper where the issue of serializing tables (and other questions covered) are relevant". We included more citations from subsequent sections into Table 1 to better summarize the papers that would appear later.
>
> 6. Personally, I think it is uninteresting to list the specific LLMs evaluated in the cited papers... The paper can also summarize the model usage by counting how many other papers surveyed used each model...
>
> Answer: We mention models because different models have different numerical feature tokenizer and pre-training strategy. We can help user to understand which base model has been studied by different tabular task. We have summarized the model usage based on the reviewer's request
>
> We have also fixed citation error.

---

### Review · Reviewer_m3jz · 2024-04-28

**Summary Of Contributions:**

This work presents a comprehensive survey on recent LLM methods for tabular data learning. The survey covers a wide range of approaches, metrics, datasets and other key aspects of the space. The paper presents a taxonomy of key techniques employed to reason over tabular data and connects the dots on an impressive range of current work.

**Audience:**

Yes

**Broader Impact Concerns:**

- This manuscript does not present any ethical concerns.

**Claims And Evidence:**

Yes

**Requested Changes:**

The paper is nice overall. I have the following suggestions for improvement and additional nuance:

**Section 3.3 : Medical Prediction**
- While EHR data is frequently treated as tabular data in many benchmarks exploring tabular learning methods (including Meditab, from a cursory examinatio), this is typically an artificially limited and overly restricted way of featurizing electronic health record (EHR) data in practice. Tabular representation methods do not reflect state-of-the-art in practice.  Since EHR data corresponds to clinical events with timestamps, there is a natural sequential, partial ordering that can be used quite effectively for deep learning based approaches that consistently outperform XGBoost methods (BERT-style): BEHRT [1], Med-BERT [2], (GPT-style): CLMBR [3], (T5-style): TransformEHR [4].
- EHR is a good setting where the tabular representation choices are more of a reflection of limited data access or otherwise missing context, I would discuss this in some capacity.

**Section 6: Limitations and future directions**
- I feel the majority of points in this section apply to LLMs broadly, not just the case with tabular reasoning.  Only (1) Hallucination (2) Categorical representation and (3) Standard benchmark were motivated with clear direct needs and opportunities specific to tabular data. Could the authors expand upon other direct connections?

**Language / Typos**
- *"It was found that PTL such as DeBERTa has been shown perform better than XGBoost in electronic health record (EHR) prediction tasks (McMaster et al., 2023)."* What is PTL?
- typo, bottom of page 13
- typo page 24 "Some pretrained model such as Wang et al. (2023c); ?"

1.  Li, Yikuan, Shishir Rao, José Roberto Ayala Solares, Abdelaali Hassaine, Rema Ramakrishnan, Dexter Canoy, Yajie Zhu, Kazem Rahimi, and Gholamreza Salimi-Khorshidi. "BEHRT: transformer for electronic health records." Scientific reports 10, no. 1 (2020): 7155.
2. Rasmy, Laila, Yang Xiang, Ziqian Xie, Cui Tao, and Degui Zhi. "Med-BERT: pretrained contextualized embeddings on large-scale structured electronic health records for disease prediction." NPJ digital medicine 4, no. 1 (2021): 86.
3. Steinberg, Ethan, Ken Jung, Jason A. Fries, Conor K. Corbin, Stephen R. Pfohl, and Nigam H. Shah. "Language models are an effective representation learning technique for electronic health record data." Journal of biomedical informatics 113 (2021): 103637.
4. Yang, Zhichao, Avijit Mitra, Weisong Liu, Dan Berlowitz, and Hong Yu. "TransformEHR: transformer-based encoder-decoder generative model to enhance prediction of disease outcomes using electronic health records." Nature Communications 14, no. 1 (2023): 7857.

**Strengths And Weaknesses:**

Strengths
- The paper covers an impressive range of papers in the tabular data space
- The sheer number of recent papers from 2022+ (the modern LLM era) speak to the need for such a survey paper
- The taxonomy and general break down of the problem is very nice and easy to follow for reader.

Weaknesses
- Some of limitations and future directions apply to LLMs generally and are not specific facets of the tabular data setting.
- For some subtopics, e.g., diffusion for data generation, the area is only mentioned broadly under "Traditional and deep learning in tabular data". When other work is discussed in comparison to these prior methods this can lead to some confusion for the reader. For example,  "However, the work lacks a comparison with diffusion-based models like TabDDPM ..."* neither TabDDPM nor the specifics of diffusion techniques have been discussed, leaving the reader to speculate on the significance of this lack of comparison.

---

> ### Author Response · Authors · 2024-05-10
> **Thank you for thoughtful feedback!**
>
> Thanks the reviewer for their thoughtful feedback! We will address them one by one.
>
> 1. Requested Change: I feel the majority of points in this section apply to LLMs broadly, not just the case with tabular reasoning. Only (1) Hallucination (2) Categorical representation and (3) Standard benchmark were motivated with clear direct needs and opportunities specific to tabular data. Could the authors expand upon other direct connections?‘
>
> Answer: Thank you for the comments. We agree that Limitations include both tabular-specific challenges and and general-LLM limitations. To address your concern, we have re-oriented our discussion on tabular data. Specifically, in the current version section 6, we clearly explain those limitations in tabular data space. For numerical representation, the tokenization of integers lacks a coherent decimal representation and some model (GPT3) split numbers into non-aligned tokens. For bias and fairness and tabular data prediction, the fairness metric gap between different subgroups is larger than that in traditional machine learning model. For model interpretability, only a few systems expose a justification of their model output, and we cannot find a standard way to interpret LLMs in tabular prediction and table understanding. For ease of use, LLM based tabular methods still require extensive preprocessing, which makes it hard to inference. For Fine-tuning strategy, existed strategy for LLM may not be fully evident in certain tabular data prediction and table understanding tasks, depending on the model used. We have also added a paragraph regarding ‘tabular-specific challenges’ and pointed out tabular features like class imbalance and order invariant is an important area for future research.
>
> 2. Weakness: For some subtopics, e.g., diffusion for data generation, the area is only mentioned broadly under "Traditional and deep learning in tabular data". When other work is discussed in comparison to these prior methods this can lead to some confusion for the reader. For example, "However, the work lacks a comparison with diffusion-based models like TabDDPM ..."* neither TabDDPM nor the specifics of diffusion techniques have been discussed, leaving the reader to speculate on the significance of this lack of comparison.
>
> Answer: The discussion on diffusion models is beyond the scope of the survey. In this survey, we mainly focus on LLM-based table synthesis. We added a new reference with a new sentence in Section 2 “A comprehensive understanding of the strengths and weaknesses of different tabular data synthesis methods can be found in [1].”
>
> [1] Du, Yuntao, and Ninghui Li. "Towards Principled Assessment of Tabular Data Synthesis Algorithms." arXiv preprint arXiv:2402.06806 (2024).
>
> 3. EHR is a good setting where the tabular representation choices are more of a reflection of limited data access or otherwise missing context, I would discuss this in some capacity.
>
> Answer: We agree with the reviewer. We mentioned relevant papers the reviewer cited, as well as discussing more about the characteristics of EHR data in section 3.4.
>
> 4. Language / Typos
> We have fixed all mentioned typos. Thank you!

---

### Decision · Action_Editor_kfxL · 2024-06-05

**Recommendation:** Accept as is

**Comment:**

The reviewers generally found this paper valuable after its revision. It's a comprehensive survey of the topic that is likely to be a valuable resource for the community at large. In some private discussion, reviewer m3jz expressed reservations about the way the paper presents claims from past work. To quote:

> In general the paper provides a surface gloss where it (uncritically) restates the claims of many papers or suggests differences that are not real. Just to zoom in on the EHR setting, where the Meditab citation was added during rebuttal. The manuscript now over-indexes into the claims made in that specific work:

>> However, these models only focused on predicting a small fraction of the International Statistical Classification of Diseases and Related Health Problems (ICD) codes. Currently, Meditab (Wang et al., 2023c) aims to create a foundation model in the medical field.

> This is not a strictly correct distinction. CLMBR, for example, predicts 65,536 codes filtered from 21 medical ontologies (not just ICD) comprising the OMOP Common Data Model's standard vocabulary (see https://huggingface.co/StanfordShahLab/clmbr-t-base). All of the comparison models (BEHRT, MedBERT, TransformEHR, CLMBR) are also foundation models. The salient difference of Meditab from the cited models, which is not clearly stated or highlighted, is that it translates codes to natural language (originally done by UniHPF (https://arxiv.org/abs/2211.08082 not cited here) to leverage pretrained LLMs.

>> The code is available for tabular prediction tasks specifically in the medical domain.

> Other cited papers also provide code and model weights in some cases.

>> On top of AUCROC, they also use a precision-recall curve (PRAUC) for evaluation. PRAUC is useful in imbalanced datasets, which is always the case for medical data.

> Other cited papers also routinely include PRAUC for the very reasons noted.

This is one example of claims from past work that could be pulled apart a bit more. I think in general, this survey could be stronger if it provided more criticism or comparison between methods. However, based on other surveys that have been accepted to TMLR and based on the review criteria, these reservations don't rise to the level of rejecting the paper. The authors should, however, consider this as they prepare the final version of the work.

**Audience:**

Yes

**Claims And Evidence:**

Yes; see below